# Brazilian Atmospheric Inventories - BRAIN: A comprehensive database of air quality in Brazil

Leonardo Hoinaski[a,*], Robson Will[b], Camilo Bastos Ribeiro[b]

[a]Department of Sanitary and Environmental Engineering, Federal University of Santa Catarina, Florianópolis, Santa Catarina, Brazil, leonardo.hoinaski@ufsc.br
[b]Graduate Program in Environmental Engineering, Federal University of Santa Catarina, Florianópolis, Santa Catarina, Brazil, robsonwillfsc@gmail.com, cb_ambiental@hotmail.com

*Correspondence to*: Leonardo Hoinaski (leonardo.hoinaski@ufsc.br)

**Abstract.** Developing air quality management systems to control the impacts of air pollution requires reliable data. However, current initiatives do not provide datasets with large spatial and temporal resolutions for developing air pollution policies in Brazil. Here, we introduce the Brazilian Atmospheric Inventories – BRAIN, the first comprehensive database of air quality and its drivers in Brazil. BRAIN encompasses hourly datasets of meteorology, emissions, and air quality. We provide gridded data in two domains, covering the Brazilian territory with 20x20 km of spatial resolution and another covering Southern Brazil with 4x4 km. The emissions dataset includes vehicular emissions derived from the Brazilian Vehicular Emissions Inventory Software (BRAVES), industrial emissions produced with local data from the Brazilian environmental agencies, biomass burning emissions from FINN - Fire Inventory from the National Center for Atmospheric Research (NCAR), and biogenic emissions from the Model of Emissions of Gases and Aerosols from Nature (MEGAN). The meteorology dataset has been derived from the Weather Research and Forecasting Model (WRF). The air quality dataset contains the surface concentration of 216 air pollutants produced from coupling meteorological and emissions datasets with the Community Multiscale Air Quality Modeling System (CMAQ). This paper describes how the datasets were produced, their limitations, and their spatiotemporal features. To evaluate the quality of the database, we compare the air quality dataset with 244 air quality monitoring stations, providing the model's performance for each measured pollutant by the monitoring stations. We present a sample of the spatial variability of emissions, meteorology, and air quality in Brazil from 2019, revealing the hotspots of emissions and air pollution issues. By making BRAIN publicly available, we aim to provide the required data for developing air quality policies on municipality and state scales, especially for not developed and data-scarce municipalities. We also envision that BRAIN has the potential to create new insights and opportunities for air pollution research in Brazil.

# 1. Introduction

It is consensus that air pollution threats public health (OECD, 2023), economic progress (OECD, 2016), and climate (USEPA, 2023a). The negative outcomes associated with air pollution are not uniform within populations and the impacts may be greater for more susceptible and exposed individuals (Makri and Stilianakis, 2008). Due to its social vulnerability and increasing emissions, developing countries urgently require reliable databases to provide information for designing air quality management systems to control air pollution (Sant'Anna et al., 2021).

Brazil has continental dimensions, is the seventh most populous country in the world, and has the 12[th] largest Gross Domestic Product (IBGE, 2023). Combining poorly planned development and the huge socioeconomic discrepancy has led to air quality impacts in Brazil. Air pollution-related problems are not only restricted to great Brazilian cities and industrialized areas. Vehicular fleet and fuel consumption have also increased in small municipalities (CEIC, 2021, MME, 2023), posing a challenge to control vehicular emissions. In preserved and rural areas, large fire emissions have occurred due to illegal deforestation and soil management (Escobar, 2019; Rajão et al., 2020).

Following practices of developed countries, Brazilian air quality policies have been enforced through legislative laws, using air quality standards as key components. However, the whole loop of the air quality management process has never been completed in Brazil. Policies are far to be efficient since the lack of air quality monitoring data in most of the country has restricted the knowledge to well-developed areas (Sant'Anna et al., 2021). Moreover, Brazilian environmental agencies have not provided enough data and guidance for permitting process. Air quality consultants still struggling to find mandatory inputs to understand and predict air quality for regulatory purposes. Efforts for permanent improvement of high spatiotemporal resolution emissions inventories, meteorological, and air quality data are needed.

An effective air quality management system must provide data to determine how much emissions reductions are needed to achieve the air quality standards (USEPA, 2023b). It requires air quality monitoring, robust and detailed emissions inventory, reliable meteorological datasets, and methodologies to adapt the state-of-the-art air quality models to Brazilian's reality. Moreover, it is crucial to undertake ongoing evaluation and fully understand the air quality problem to design and implement the programs for pollution control. Currently, available initiatives including reanalysis and satellite products are still not providing datasets with large spatial and temporal resolutions for developing air pollution policies in Brazil. Global reanalysis such as Copernicus Atmospheric Monitoring Service (CAMS) (Inness et al., 2018) and the second version of Modern-Era Retrospective Analysis for Research and Applications (MERRA-2) (GMAO, 2015a,b) can provide estimates of air pollutants by combining chemical transport models (CTMs) with satellite and ground-based observations and physical information, assimilating data to constrain the results. However, the currently available reanalysis products do not provide data with high spatial resolution (up to 0.75° × 0.75° and 0.5° x 0.625°) and could be biased to represent local and regional air quality (Arfan Ali et al., 2022). Moreover, they provide data only for a small list of air pollutants. Satellite-based products such as Sentinel-5P TROPOMI (Veefkind et al., 2012) and Moderate-Resolution Imaging Spectroradiometer (MODIS) (Levy et al., 2015; Platnick et al., 2015) are still challenging due to their low temporal resolution, data gaps due to cloud coverage, and uncertainties (Shin et al., 2019). Besides, satellite relies on total tropospheric column measurements which do not represent surface concentrations (Shin et al., 2019).

In this article, we present the Brazilian Atmospheric Inventories (BRAIN), the first comprehensive database to elaborate air quality management systems in Brazil. BRAIN combines local inventories, consolidated datasets, and the usage of internationally recommended models to provide hourly emissions, meteorological, and air quality data covering the entire country.

## 2. BRAIN Database

BRAIN contains three types of hourly datasets: emissions, meteorology, and air quality. The emissions inventories include vehicular, industrial, biogenic, and biomass-burning emissions. We provide meteorological data derived from Weather Research and Forecasting (WRF) model. Coupling emissions, WRF, and the Community Multiscale Air Quality Modeling System (CMAQ) version 5.3.2, we provide air quality gridded data. All datasets are available on two spatial resolutions, the largest (Figure SM1 – d01) covers the entire country, while the smallest covers southern Brazil (Figure SM1 – d02). The

BRAIN datasets in d01 are freely available at https://doi.org/10.57760/sciencedb.09858 (Hoinaski et al., 2023a), https://doi.org/10.57760/sciencedb.09857 (Hoinaski and Will, 2023a), and https://doi.org/10.57760/sciencedb.09859 (Hoinaski and Will, 2023b). The BRAIN datasets in d02 are available at https://doi.org/10.57760/sciencedb.09886 (Hoinaski et al., 2023b), https://doi.org/10.57760/sciencedb.09885 (Hoinaski and Will, 2023c), and https://doi.org/10.57760/sciencedb.09884 (Hoinaski and Will, 2023d). The Federal University of Santa Catarina (UFSC)

institutional repository https://brain.ens.ufsc.br/ and the web platform https://hoinaski.prof.ufsc.br/BRAIN/ serve the BRAIN database from 2019. We envision making available more detailed datasets for other Brazilian regions, especially in the Southeast where the anthropogenic emission effects are more prominent. Future versions will also provide more detailed modeling outputs to properly cover medium- and small-sized cities.

BRAIN intends to fill the gaps in those cases where adequately representative monitoring data to characterize the air quality

is not available. BRAIN would be useful to provide background concentrations for a good procedure for licensing new sources of air pollution.

### 2.1 Emissions inventory

BRAIN emissions inventory allows the spatiotemporal analysis of vehicular, industrial, biomass burning, and biogenic emissions in Brazil. The present version of this database does not account for other South American countries emissions, apart

from biomass burning and biogenic sources. We envision to implement other sources and a more detailed emissions from other South American countries in future version. Figure 1 presents a sample of the inventory, showing the annual Carbon Monoxide (CO) emissions by source. Table SM2 summarizes the species in each emission source inventory. More information on each emissions dataset can be found in sections 2.1.1 to 2.1.5.

We observed the higher vehicular emissions rates of CO in urban areas with large population and vehicle fleet densities, mainly

in the South and Southeast (Figure 1a). High industrial emission rates have been detected in the Brazilian regions with large stationary sources such as refining units, thermoelectric power plants, cement, and paper industries (Figure 1b) (Kawashima et al., 2020). In general, the North concentrates higher biogenic and fire emissions. While the hotspots of biogenic emissions are predominately in the extreme North, the hotspots of fire emissions turn up in Mid-West, North, and South regions, as well as in the Brazilian west border (Figure 1c-d).

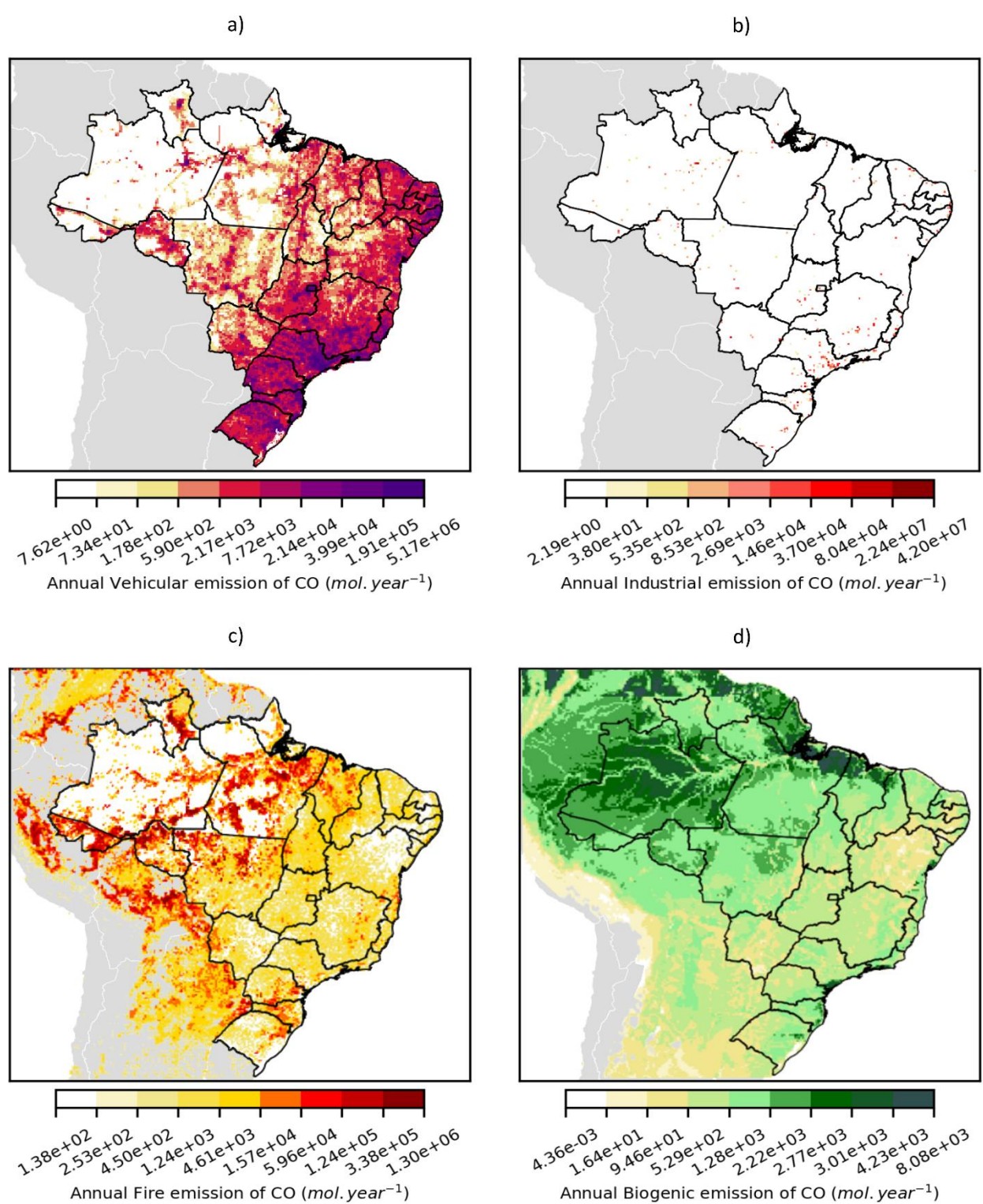

**Figure 1. Spatial distribution of CO emissions from a) vehicles, b) industries, c) biomass burning, d) biogenic provided by BRAIN.**

### 2.1.1 Vehicular emissions

BRAIN uses the multispecies and high-spatiotemporal-resolution database vehicular emissions from Brazilian Vehicular Emission Inventory Software – BRAVES (Hoinaski et al., 2022; Vasques and Hoinaski, 2021). BRAVES database disaggregates municipality aggregated emissions using the road density approach and temporal disaggregation based on vehicular flow profiles. SPECIATE 5.1 (USEPA, 2020; Eyth et al., 2020) from United States Environmental Protection Agency (USEPA) https://www.epa.gov/air-emissions-modeling/speciate) speciates conventional pollutants in 41 species. BRAVES

considers local studies (Nogueira et al., 2015) and data from Companhia Ambiental do Estado de São Paulo (CETESB) (https://cetesb.sp.gov.br/veicular/relatorios-e-publicacoes/) to speciate acetaldehydes, formaldehyde, ethanol, and aldehydes since to account for biofuels particularities in Brazil.

In BRAVES, vehicular activity is defined by fuel consumption in each municipality using data provided by the Brazilian National Agency for Oil, Natural Gas and Biofuel (ANP) (https://www.gov.br/anp/pt-br/centrais-de-conteudo/dados-abertos/vendas-de-derivados-de-petroleo-e-biocombustiveis). A fraction of fuel consumed by road transportation is based on data from National Energy Balance (BEN) (https://www.epe.gov.br/pt/publicacoes-dados-abertos/publicacoes/balanco-energetico-nacional-ben), and MMA, (2014). BRAVES calculates weighted Emission Factor (EF) to address the effect of the fleet composition in terms of category, model-year, and fuel utilization.

Vasques and Hoinaski, (2021) compared BRAVES with different vehicular emission inventories, from a local to national scale. On a national scale, vehicular emission rates from BRAVES underestimate Emission Database for Global Atmospheric Research (EDGAR) and are slightly higher for CO (14%) and Non-Methane Volatile Organic Compounds (NMVOC) (9%) compared with the national inventory from Ministério do Meio Ambiente (MMA). The differences between estimates from BRAVES and well-developed state inventories vary from –1% to 35% in São Paulo and from –2% to 52% in Minas Gerais. In addition, a relatively small bias between BRAVES and Vehicular Emission Inventory (VEIN) was observed in São Paulo and Vale do Paraiba (Vasques and Hoinaski, 2021).

### 2.1.2 Industrial emissions

We derived the industrial emissions inventory by combining data from state environmental agencies of Espírito Santo, Minas Gerais, and Santa Catarina. The emission rates of point sources from Espírito Santo and Minas Gerais are publicly provided by Instituto de Meio Ambiente e Recursos Hídricos do Espírito Santo (IEMA-ES) (https://iema.es.gov.br/qualidadedoar/inventariodefontes) and Fundação Estadual de Meio Ambiente (FEAM) (http://www.feam.br/qualidade-do-ar/emissao-de-fontes-fixas). Data from IEMA-ES contains emissions from Vitória Metropolitan Region from 2015, compiling measurements from regulatory procedures and emissions estimates. We did not convert the emissions inventory to the current modeling year since the data is not continuously updated. Therefore, we assumed that all emissions from these multiple sources occurred in 2019.

In Santa Catarina, industrial emission data has been provided by Instituto de Meio Ambiente (IMA) (https://www.ima.sc.gov.br/index.php). These data are collected in the licensing process of potentially polluting industries. The base year of emission rates varies according to the availability. Summary information about the industrial sector types, the number of industries, and the respective emission rates in Santa Catarina can be found in Hoinaski et al., (2020) and at https://github.com/leohoinaski/IND_Inventory/blob/main/Inputs/BR_Ind.xlsx. Emissions from large stationary sources (refining units, thermoelectric power plants, cement, and paper industries) provided by Kawashima et al., (2020) have been included when not encountered in the environmental agencies' inventories.

We chemically speciated the industrial emission rates adopting the following steps: i) grouping each point source using the same categories as in Emission Database for Global Atmospheric Research (EDGAR) (Crippa et al., 2018) and Intergovernmental Panel on Climate Change (IPCC) industrial segments; ii) selecting compatible profiles in SPECIATE 5.1 for each group (Eyth et al., 2020); iii) averaging the speciation factor for by group and pollutant, and iv) applying the speciation factor for the targeted pollutant (PM, NOx, VOCs). The SPECIATE 5.1 profiles used in this work are listed in https://github.com/leohoinaski/IND_Inventory/tree/main/IndustrialSpeciation. The speciation factor by industrial group and pollutant are available at: https://github.com/leohoinaski/IND_Inventory/blob/main/IndustrialSpeciation/IND_speciation.csv.

We also vertically allocate the industrial emissions according to the plume's effective height, estimated by the sum of the geometric height and superelevation of the plume. The plume superelevation was estimated by the Briggs method (Briggs,

1975, 1969). The initial vertical distribution of the plume has been estimated by disaggregating the emissions using a Gaussian approach, as proposed in the Sparse Matrix Operator Kernel Emissions (SMOKE) model (Bieser et al., 2011; Gordon et al., 2018; Guevara et al., 2014). Python code to estimate the plume's effective height and the initial vertical disaggregation of industrial emissions is available at https://github.com/leohoinaski/IND_Inventory.

### 2.1.3 Biomass burning emissions

Fire Inventory from NCAR (FINN) version 1.5 (Wiedinmyer et al., 2011) provides data from biomass burning emissions in BRAIN. FINN outputs contain daily emissions of trace gas and particle emissions from wildfires, agricultural fires, and prescribed burnings and do not include biofuel use and trash burning. Datasets have 1km of spatial resolution and are available at https://www.acom.ucar.edu/Data/fire/.

Since CMAQ requires hourly emissions, a Python code (https://github.com/barronh/finn2cmaq) temporally disaggregates daily emissions into hourly emissions. The same code vertically splits the fire emissions to consider the plume rise effect and represents the vertical distribution (Henderson, 2022), converting text files into hourly 3D netCDF files.

Pereira et al., (2016) suggest that fire emissions estimated by FINN are strongly related to deforestation in many Brazilian regions. FINN estimates have a high correlation both with the Brazilian Biomass Burning Emission Model (3BEM) (0.86) and Global Fire Assimilation System (GFAS) (0.84). The emissions estimated from FINN commonly overestimated other biomass burning emission inventories. An overestimation also occurs when FINN is used in air quality models and compared with observations. However, the use of FINN as input in air quality models can capture the temporal variability of pollutants emitted by biomass burning (Vongruang et al., 2017).

We have implemented the FINNv1.5 in this first version of BRAIN. However, FINN version 2.5 (Wiedinmyer et al., 2023) will be included in our emissions inventory in future work, which uses an updated algorithm for determining fire size based on MODIS and Visible Infrared Imaging Radiometer Suite (VIIRS) satellite instruments. We also provide data from 2020 with the same modeling grid upgrading to FINN v2.5.

### 2.1.4 Biogenic emissions

We derived the biogenic emissions using the Model of Emissions of Gases and Aerosols from Nature (MEGAN) version 3.2 (Guenther et al., 2012; Silva et al., 2020). MEGAN is based on the leaf area index and plant functional groups. The model estimates emissions of gases and aerosols for different meteorological conditions and land cover types (Guenther et al., 2012). The leaf-level temperature and photosynthetically active radiation, as well as the vegetative stress conditions implemented in MEGAN, provide more physically realistic parameterizations for biosphere-atmosphere interactions (Silva et al., 2020). Input datasets, emission factor processors, and emission estimation module are available at https://bai.ess.uci.edu/megan/data-and-code. Data from WRF and Meteorology-Chemistry Interface Processor (MCIP) have been used in MEGAN simulations.

MEGAN is commonly adopted to estimate emissions from biogenic fluxes, which is an important input for air quality modeling in many regions worldwide (Hogrefe et al., 2011; Kitagawa et al., 2022; Kota et al., 2015). Although MEGAN overestimates nighttime biogenic fluxes, the modeled emissions are correlated with measurements in Amazon, both during wet and dry seasons. The model is capable to capture relatively well the seasonal variability of important organic pollutants in tropical forests (Sindelarova et al., 2014).

### 2.1.5 Sea spray aerosol emissions

Sea spray aerosol (SSA) is an important source of particles in the atmosphere. Due to its properties, SSA influences gas-particle partitioning in coastal environments (Gantt et al., 2015). SSA has been implemented in CMAQ as an inline source and requires

the input of an ocean mask file (OCEAN) to identify the fractional coverage in each model grid cell allocated to the open ocean (OPEN) or surf zone (SURF). CMAQ uses this coverage information to calculate sea spray emission fluxes from the model's grid cells (USEPA, 2022). Detailed information on the mechanism of sea spray aerosol emissions and its implementation on CMAQ can be found in (Gantt et al., 2015).

We provide a Python code (https://github.com/leohoinaski/CMAQrunner/blob/master/hoinaskiSURFZONEv2.py) to reproduce the OCEAN time-independent Input/Output Applications Programming Interface (I/O API) (https://www.cmascenter.org/ioapi/) file ready to use in CMAQ. This code uses a shoreline Environmental Systems Research Institute (ESRI) shapefile from National Oceanic and Atmospheric Administration (NOAA) available at https://www.ngdc.noaa.gov/mgg/shorelines/.

## 2.2 Meteorology

WRF model has been used in this work to produce inputs for CMAQ and for meteorology characterization in Brazil. We provide hourly simulation in netCDF files. WRF has been set up to reproduce 36 hours simulations, where the initial 12 hours have been dedicated to model stabilization, which are excluded from the analysis. Thirty-three vertical levels have been employed, spaced at 50 hPa intervals. The parameterizations used in this work are described in SM3. The remaining vertical levels followed a hybrid modeling scheme, accounting for terrain in the lower layers and gradually minimizing its influence in the higher levels. Details of WRF outputs can be found in SM4.

Global Forecast System (GFS) from the National Center for Atmospheric Research (NCAR) provided inputs with spatial resolution of 0.25° x 0.25° and a temporal resolution of six hours for the WRF simulations (Skamarock et al., 2008). Land use data and classification parameters from the United States Geological Survey's (USGS) Moderate Resolution Imaging Spectroradiometer (MODIS).

The Brazilian regions (North, North-East, Mid-West, South-East, and South) encompass three distinct climatic zones, namely the equatorial, tropical, and subtropical zones. The climatic diversity in Brazil is also shaped by topographical variations, landscape/vegetation, and the coastal areas. The temperature in Brazil follows a latitudinal pattern, increasing from South to North (Figure 2e). The highest average temperatures are observed in the Amazon region, matching the historic data (Cavalcanti, 2016). The South region exhibits the lowest average temperatures, which is also consistent with historical data (Cavalcanti, 2016).

The highest values of atmospheric pressure occurred in the North region and the extreme South of the country, and the lowest values were between the South-East and South regions (Figure 2a). The planetary boundary layer height (PBLH) reaches the highest levels in the North-East region and the lowest in the South and South-East coast (Figure 2b). The highest values of wind speed occurred in part of the North and South region. The Amazon region presented the lowest values of surface wind speed (Figure 2f).

Humidity and precipitation exhibit similar patterns in the North and Northeast regions (Figure 2 cd), due to the trade winds that transport moisture from the tropical Atlantic (Mendonça and Danni-Oliveira, 2017). Except for the coast, the North-East region is characterized by low humidity and drought during half of the year. The South and South-East regions have well-distributed rainfall throughout the year, as well as intermediate levels of humidity, except for the northern coast of the South region, which have an elevated level of precipitation and humidity throughout the year.

The WRF model demonstrated the ability to reproduce diurnal and seasonal variability of winds in the Brazilian North-East region (Souza et al., 2022a), although it underestimated the height of the planetary boundary layer (PBLH) by up to 20%, as well as the temperature and humidity at 4°C and 15%, respectively. Pedruzzi et al. (2022) tested several model configurations, including an alternative land use scheme, and found a WRF tendency to overestimate temperature and humidity in the Brazilian South-East region. Macedo et al., 2016 also evaluated the model's ability to predict extreme precipitation events. Although the

WRF reasonably predict the main meteorological aspects of the Brazilian South region, the precipitation extremes were underestimated. A wind mapping study (Souza et al., 2022b) using WRF indicated that the average errors presented by the model in Brazil are minor, with an average bias of 2m/s at 200m in wind intensity, and errors at temperatures of 2°C and humidity of approximately 10%. Winds at lower levels tended to be overestimated, whereas PBLH was generally underestimated during the day.

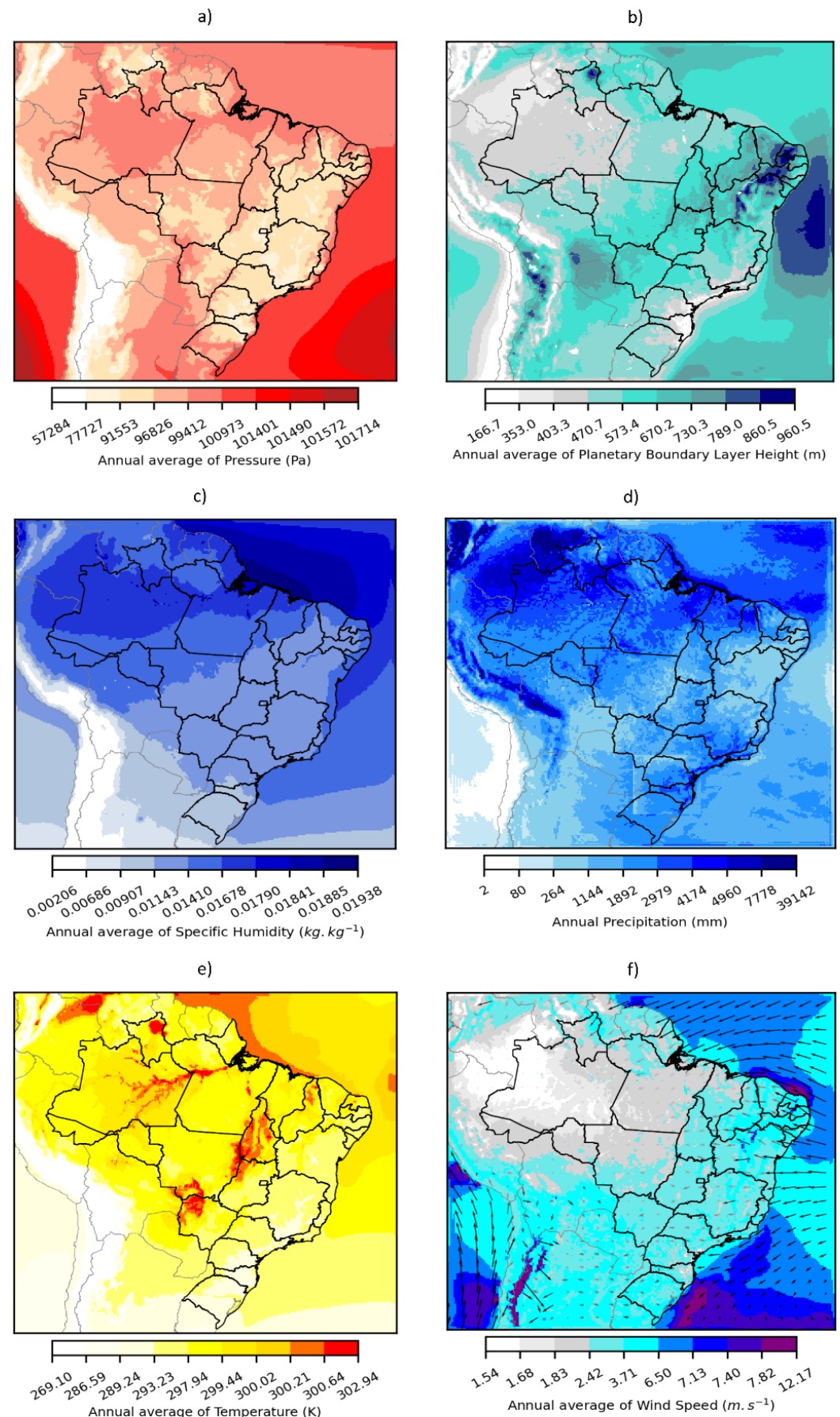

**Figure 2. Annual average of meteorological variables in 2019, simulated by the WRF with 20 x 20 km resolution. (a) Atmospheric pressure, (b) Planetary boundary layer height, (c) Specific humidity, (d) Annual accumulated precipitation, (e) Temperature, (f) Wind intensity and direction. All variables are annual averages except for precipitation, which represents the annual accumulated total.**

**2.3 Air quality**

We coupled emissions inventories, WRF, and CMAQ to produce the BRAIN air quality dataset for Brazil. CMAQ version
5.3.2 was set up using the third version of the Carbon Bond 6 chemical mechanism (cb6r3_ae7_aq) (Yarwood et al., 2010; Emery et al., 2015) with AERO7 treatment of Secondary Organic Aerosol for standard cloud chemistry (Wyat Appel et al., 2021). Other model's configuration used in this work can be found in SM5 and https://github.com/leohoinaski/CMAQrunner. The pollutant list in CMAQ outputs containing 216 species can be found in SM6.

The CMAQ standard profile of boundary conditions is used in the larger domain (d01), which provides the boundary conditions
for the smaller one (d02). Further improvements of the database could include the boundary conditions derived from the GEOS-Chem model (Bey et al., 2001) (https://geoschem.github.io/) or other better alternatives for the largest domain. The simulations have 24 hours length and time step interval of 1 hour. The last hour of the previous simulation has been set up as the initial condition of the next one. We used the standard profile for the first hour of the first simulation (00:00:00 01-01-2019). The figures with the spatial distribution and violations of criteria pollutants can be found in SM7. SM8 also presents
the time-series of criteria pollutants in Brazilian cities.

Using BRAIN air quality dataset, we can observe the highest concentrations of $NO_2$ (Figure 3a-b), $O_3$ (Figure 3c-d), and $PM_{10}$ (Figure 3e-f) in South-East and South Brazil. The concentration violates the World Health Organization (WHO) air quality standards in multiple locations all over the country for $O_3$, while for $NO_2$ and $PM_{10}$ it occurred mostly in South-East and South Brazil.

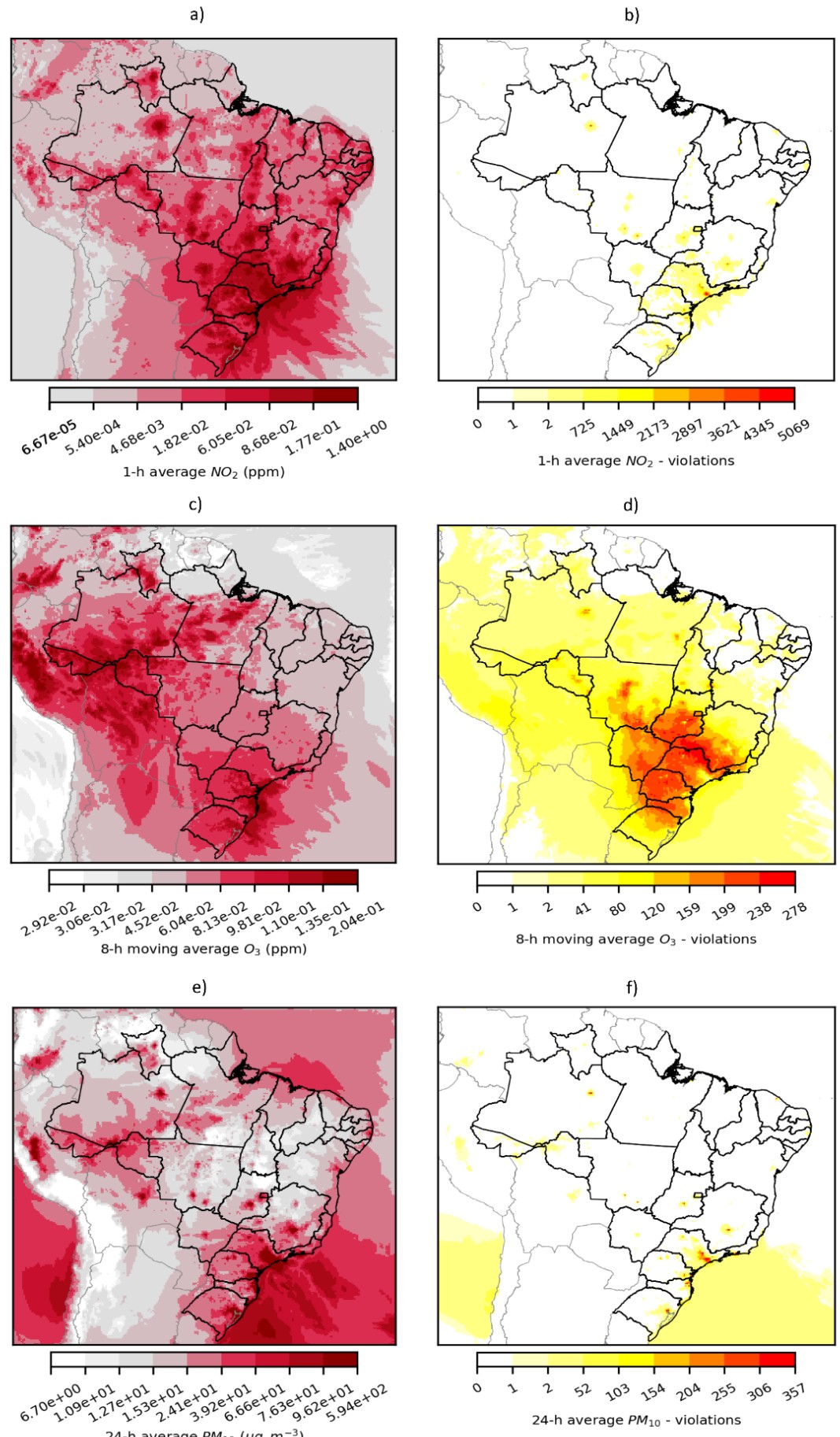

**Figure 3. Spatial distribution of air pollutant concentration (a, c, e) and number of violations of air quality standards (b, d, f) for NO₂ (a-b), O₃ (c-d), and PM₁₀ (e-f).**

### 2.3.1 Models' performance

We sampled pixels around the monitoring station using a buffer of 0.5° degrees to calculate the Spearman rank, bias, Root Mean Squared Error (RMSE), and Mean Absolute Error (MAE) of the sampled pixels. We selected the highest Spearman rank of each pixel to demonstrate the model's performance in Figures 4 and 5. SM10 presents the boxplots with overall statistical metrics for all stations. SM11 presents statistical metrics by a monitoring station and pollutant, considering the pixel with the highest Spearman rank around each monitoring station. SM12 presents the scatterplots comparing BRAIN air quality dataset and observations of each monitoring station. We used the simulations with domain d01 in the statistical analysis.

We observed the highest Spearman rank (0.72) in the state of São Paulo for $O_3$ concentration. Bias analysis revealed an underestimation in São Paulo metropolitan area, while an overestimation occurred in Minas Gerais, Santa Catarina, Rio Grande do Sul, and the interior of São Paulo. In the North-East and the state of Espírito Santo, bias is closer to zero. In Rio de Janeiro, the model over and underestimated the observations. Regarding RMSE and MAE, the model performed better in coastal areas (maps in Figure 4).

Comparing the states with air quality monitoring stations, the Spearman correlation of the $O_3$ dataset from BRAIN is higher in São Paulo, Minas Gerais, and Rio de Janeiro. However, these states also have a higher range of bias values, which could be negative and positive in São Paulo and Rio de Janeiro, and only positive in Minas Gerais (boxplots in Figure 4).

The heterogeneity in the stations' type and the insufficient spatial representativeness of observations in the Brazilian states must be considered while evaluating the model performance. According to the IEMA (2022), the strategic planning for the implementation of air quality monitoring stations, the financing and political efforts, and the technical characteristics (from installation to calibration and maintenance) vary significantly between Brazilian states. The lack of data quality assurance may compromise the credibility of the available air quality observations in Brazil.

BRAIN well reproduced the concentrations in moderately urbanized areas, such as Limeira and Piracicaba (Figures in SM12). The database reached moderate performance in highly urbanized areas such as Copacabana/RJ and at Marginal Tietê in the megacity of São Paulo (Figure in SM12). Regarding the temporal profiles of $O_3$ and $PM_{10}$, the seasonal and daily profiles are captured for both modeled pollutants, showing a suitable fit with the observation at Limeira and Pecém Industrial and Port Complex (CIPP) air quality monitoring stations (Figure 5). It reveals that the database can capture temporal patterns of air pollutant concentrations in urbanized and industrialized areas.

Figures with statistical metrics for other pollutants can be found in SM13. Figures of modeled and observed timeseries for all monitoring stations can be found in SM14.

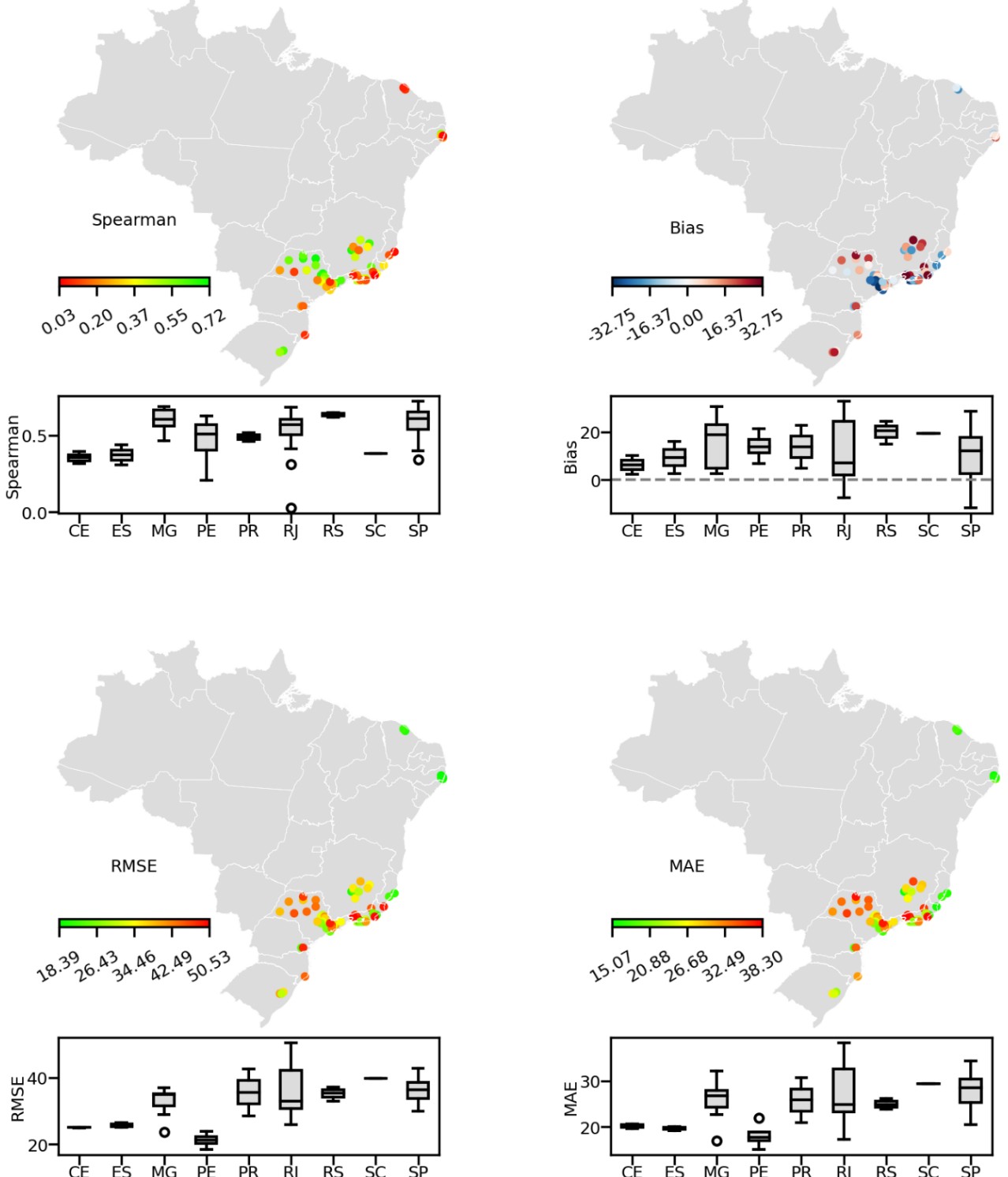

**Figure 4. Spearman rank, bias, Root Mean Squared Error (RMSE), and Mean Absolute Error (MAE) of O$_3$ dataset from BRAIN *vs* observed values. Boxplots of statistical metric by Brazilian state (considering only states with monitoring stations with representative data in 2019).**

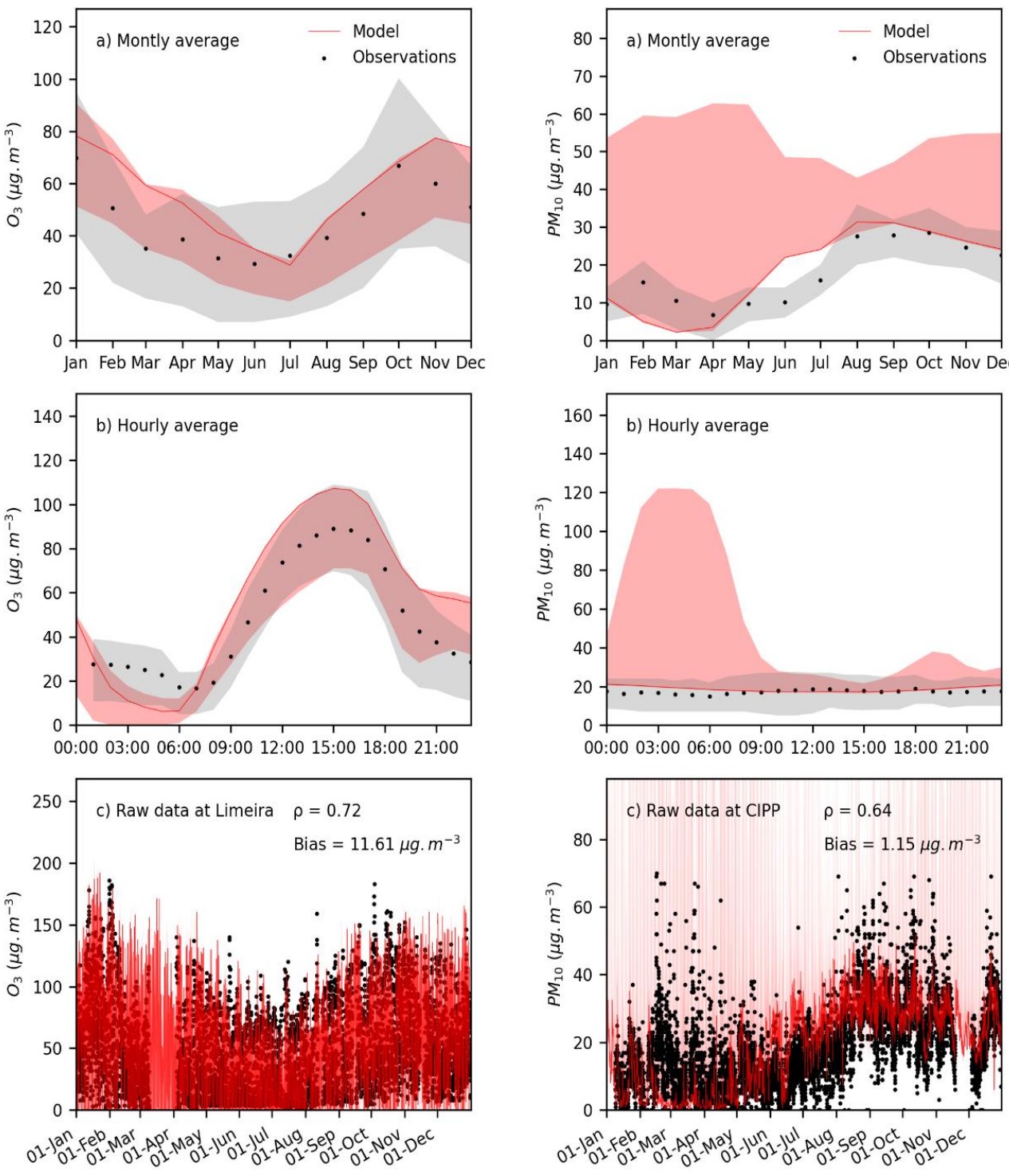

**Figure 5. Hourly (a), daily (b), and monthly (c) time-series of O₃ and PM₁₀ modeled and measured in Limeira (left) and CIPP (right) monitoring stations.**

Overall, the average concentrations are well simulated by CMAQ in BRAIN, with fair to good correlations (up to ~0.7) between modeling and local measurement in São Paulo. Similar results have been reported by Albuquerque et al., (2018). Kitagawa et al., (2021) simulated $PM_{2.5}$ in a Brazilian coastal-urban area and showed that the CMAQ results commonly overestimated the observations, which agrees with the BRAIN air quality dataset. In another comparison between observations

and CMAQ simulations (Kitagawa et al., 2022), the model overestimated the PM and $NO_2$ concentrations in the Metropolitan Region of Vitoria (MRV) and underestimated $O_3$. The authors suggest that the CMAQ simulations are suitable over the MRV, even though the model could not capture some local variabilities of air pollutant concentrations. It is already reported that the

short-time abrupt variations are difficult to reproduce by air quality models (Albuquerque et al., 2018). The complex task of predicting air quality is associated with multiple error factors, including the lack of emissions inventory, meteorology parameterizations, initial and boundary conditions, chemical mechanisms, numerical routines, etc. (Cheng et al., 2019; Albuquerque et al., 2018; Park et al., 2006; Pedruzzi et al., 2019).

We analyzed the performances of 4x4 km simulations for CO, $NO_2$, $O_3$, and $SO_2$ drawing a buffer of 0.5° degrees around monitoring station positions in southern Brazil. Our findings indicated higher Spearman values for the spatial resolution of 20x20 km for CO, $O_3$, and $SO_2$. Specifically, for $O_3$, the best result at 20x20 km was 0.76, whereas the same point at 4x4 km resolution showed a correlation of 0.46. This pattern was also observed for CO, with the best result at 20x20 km being 0.47 for Spearman and 0.23 at the same point at 4x4 km resolution. The smallest differences in Spearman rank were observed for $SO_2$ (0.22: 20x20, 0.19: 4x4). Even though improving spatial resolution did not increase the correlation with measured data, we found best results for Bias, RMSE, and MAE for almost all pollutants at a 4x4 km resolution, except for CO. Please refer to SM15 for the complete statistical analysis of 4x4 km simulations.

BRAIN captures seasonal patterns and the absolute magnitude of $PM_{2.5}$ in the Northwest of the Amazonas state (near the Amazon Tall Tower Observatory -ATTO) presented by Artaxo et al., (2013). It shows that our database can reproduce the concentrations in background areas (far from highly urbanized centers). Comparing BRAIN with observations at heavy biomass burning impacted sites in south-western Amazonia (Porto Velho) (Artaxo et al., 2013) revealed that BRAIN can capture seasonal variations caused by wet and dry seasons and the magnitude of average and peak concentrations. However, BRAIN $PM_{2.5}$ estimates are closer to the coarse mode of the time series, rather than the fine mode shown in Artaxo et al., 2013. Even though BRAIN has captured the $O_3$ pattern observed by Artaxo et al. (2013), the estimates are around 2.7 higher than the observations in the dry season and a factor of 2 higher for the wet season. It is worth mentioning that BRAIN uses 2019 data, while Artaxo et al. (2013) consists of a sampling campaign from 2008 to 2012.

BRAIN has a similar spatial pattern compared with MERRA-2 (GMAO, 2015a b), capturing hotspots in higher populated areas located in the Southeast, South, and Mid-West. In the Amazon region, BRAIN can also capture hotspots similar to MERRA-2 (Figure 6). BRAIN estimates for carbon monoxide are lower than MERRA-2, except for the South region and some urban centers in the Southeast and Midwest (Figure 6). Carbon monoxide concentrations estimated by BRAIN are moderately correlated with MERRA-2 mainly in the South (0.57) and Southeast (0.55), while in the Midwest, North, and Northeast the correlation is weaker (Figure 7). Compared with the consolidated MERRA-2 database, BRAIN has the advantage since it uses local and more refined information and provides data in higher spatial resolution for multiple species. We provide a detailed comparison between MERRA-2 and BRAIN datasets for $PM_{2.5}$, $SO_2$, $O_3$, and CO in SM16.

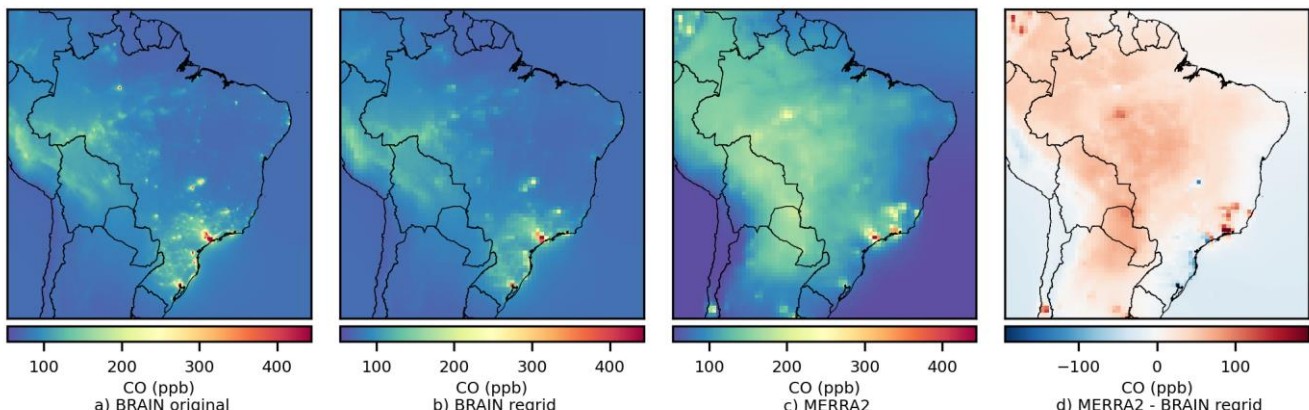

**Figure 6. Annual average concentration of CO from BRAIN original resolution (a), BRAIN regridded to MERRA-2 resolution (b), MERRA-2 (c), and difference between MERRA-2 and BRAIN (d).**

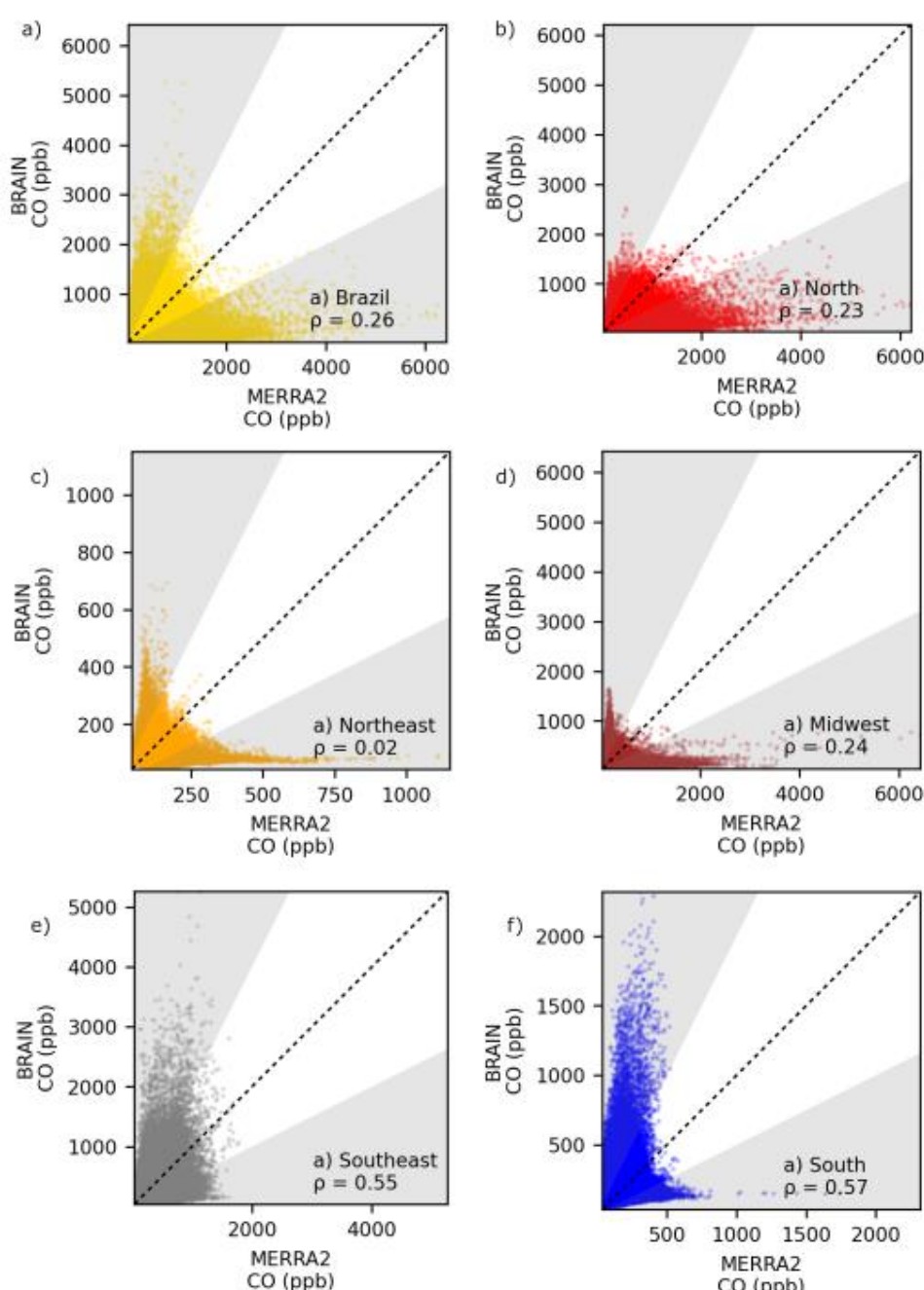

**Figure 7. Concentration of CO from BRAIN vs MERRA-2 in Brazil (a), North Brazil (b), Northeast Brazil (c), Midwest Brazil (d), Southeast Brazil (e), South Brazil (f).**

We also compare our database with Sentinel-5P TROPOMI (Veefkind et al., 2012) data to demonstrate BRAIN's ability to
capture the spatiotemporal variability of air pollutants in unmonitored areas (Figure 8). We spatially realign Sentinel-5P
TROPOMI products in the BRAIN resolution (20x20km), using data from the NASA Goddard Earth Sciences Data and
Information Services Center (GES-DISC) (https://disc.gsfc.nasa.gov/). We merged all layers of the same day and interpolated
to match BRAIN resolution. We computed the daily averages for both datasets. In this evaluation, we must consider the
differences between the datasets, since Sentinel-5P TROPOMI relies on tropospheric column measurements and BRAIN
surface concentrations. BRAIN captured the hotspots of CO and $NO_2$ similar to Sentinel-5P TROPOMI products, especially
in Southeast Brazil. However, the hotspots of CO are dislocated towards the ocean in Sentinel-5P TROPOMI. $NO_2$ estimates
from BRAIN present a higher number of hotspots. We emphasize that surface concentrations data are more suitable than
tropospheric column data in representing air quality. In this analysis, we removed negative values from Sentinel-5P TROPOMI
products since they represent low quality measurements (Eskes et al., 2022).

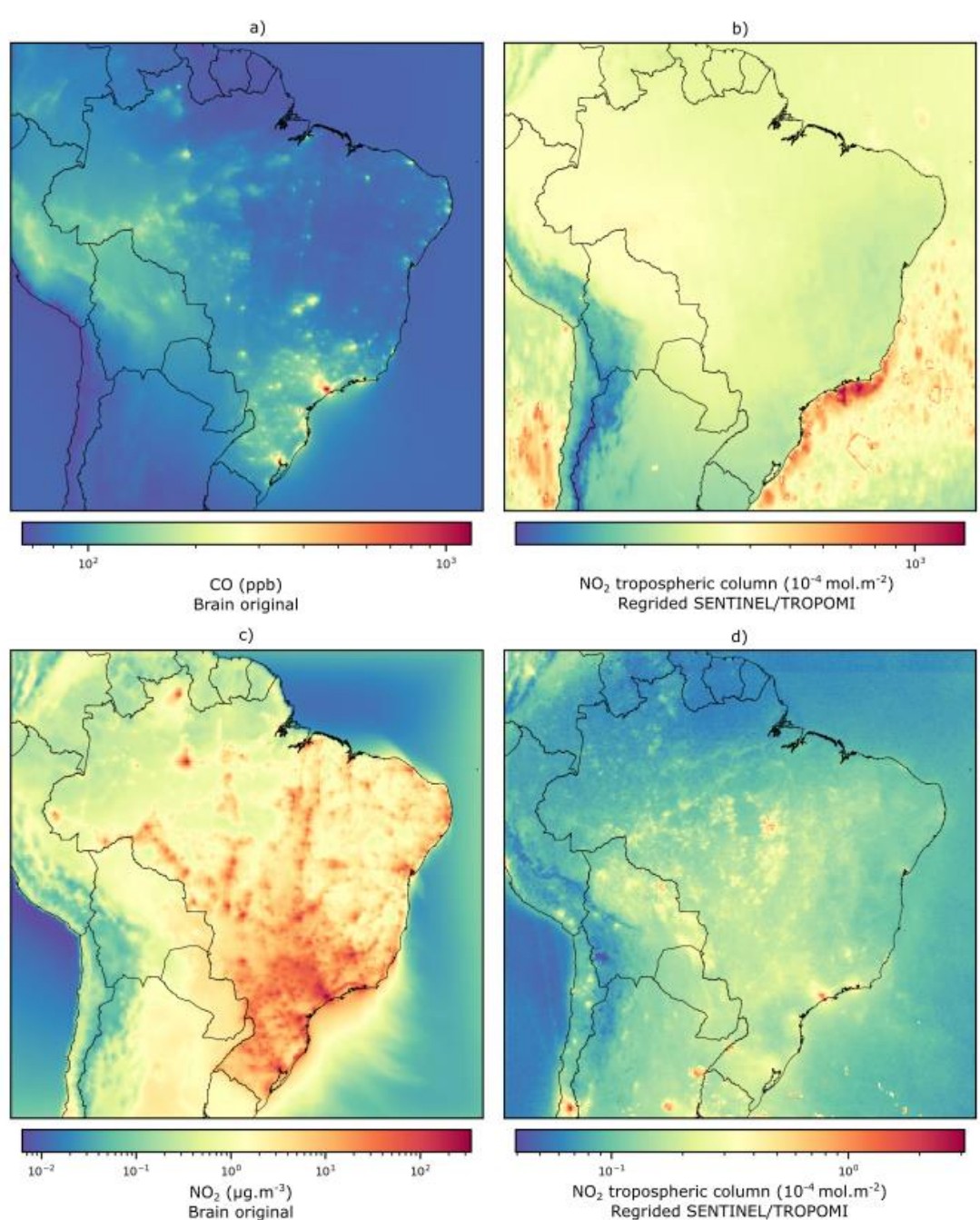

**Figure 8. Annual average concentration of CO and NO₂ from BRAIN original resolution (a, c), Sentinel-5P TROPOMI spatially aligned to BRAIN resolution (b, d).**

When we compared CO daily datasets from BRAIN and Sentinel-5P TROPOMI by Brazilian regions, we observed a moderate correlation in North (0.41), Midwest (0.32), and South (0.3). This analysis shows that BRAIN can reasonably detect temporal and spatial patterns of air pollutants. The complete comparison of CO and NO₂ from Sentinel-5P TROPOMI and BRAIN can be found in SM17.

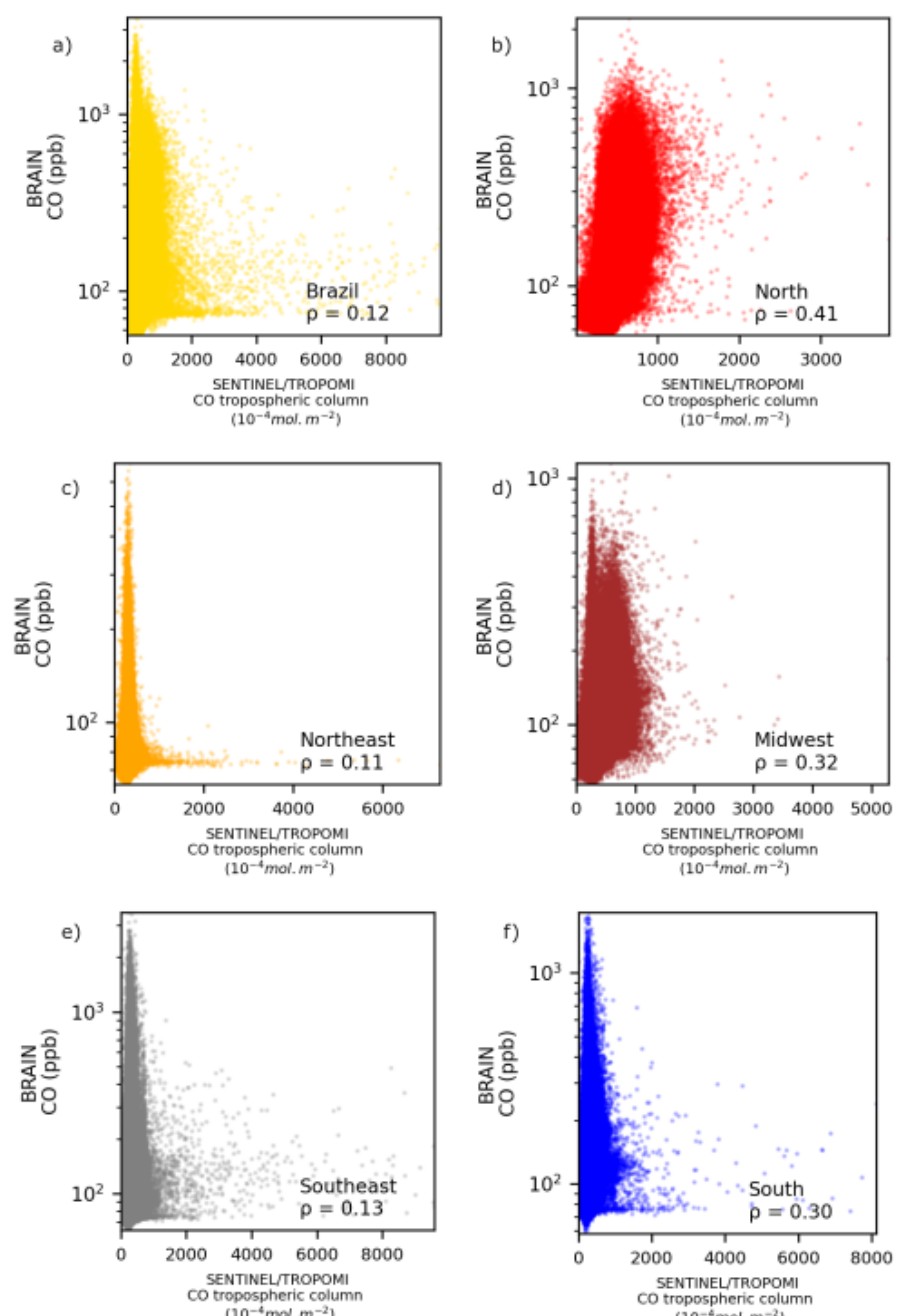

**Figure 9. Concentration of CO from BRAIN vs Sentinel-5P TROPOMI in Brazil (a), North Brazil (b), Northeast Brazil (c), Midwest Brazil (d), Southeast Brazil (e), South Brazil (f).**

We highlight that BRAIN, MERRA-2, and Sentinel-5P TROPOMI can capture similar temporal patterns of air pollutant concentrations in heavy biomass-burning impacted sites such as Porto Velho in Rondônia (Figure 10) and urban areas such as São Paulo. We provide time series (minimum-maximum and average) of BRAIN, MERRA-2, and Sentinel-5P TROPOMI data spatially averaged within Brazilian capitals in SM18. SM19 contains time series (only average) of BRAIN data in Brazilian capitals.

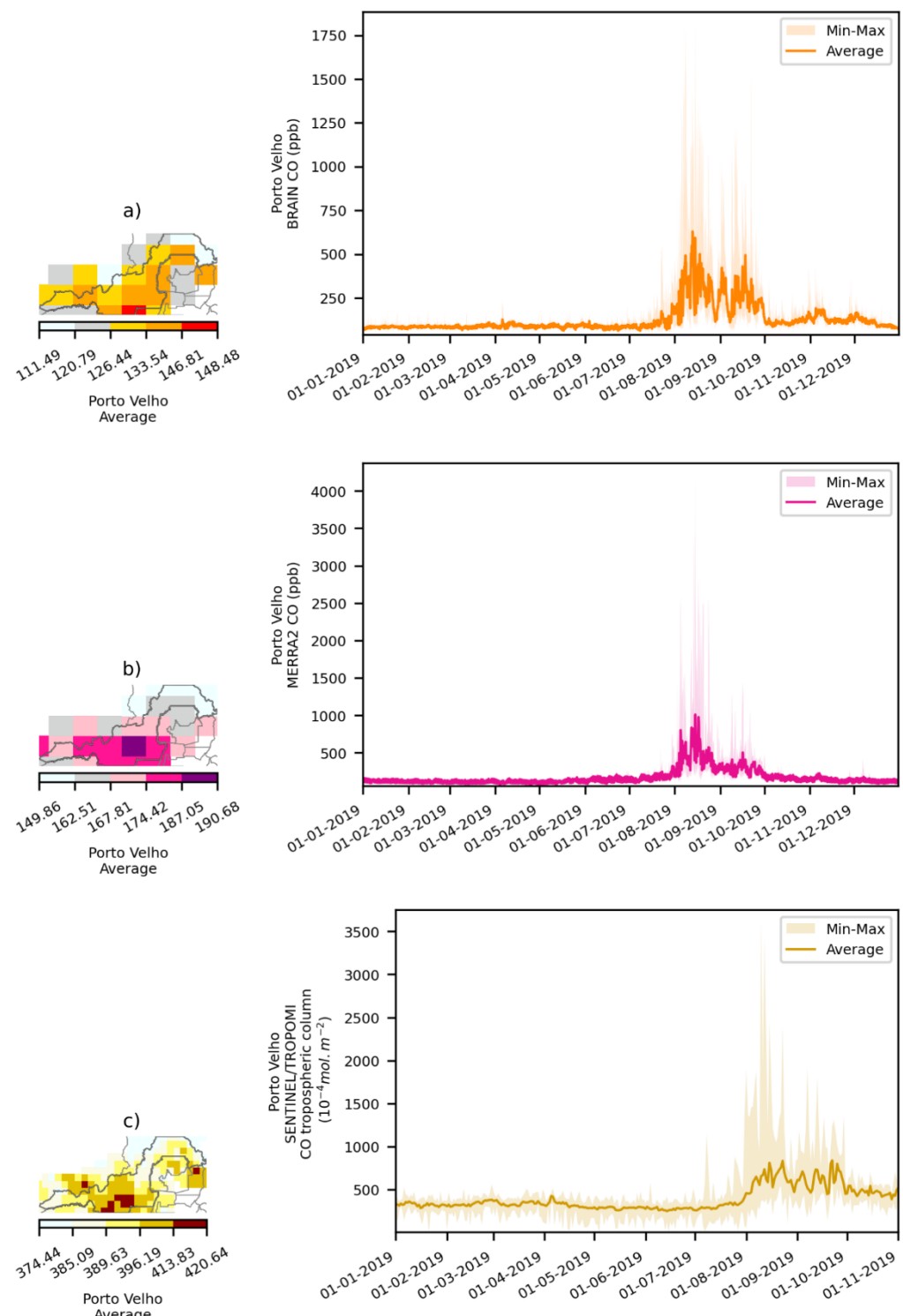

Figure 10. Annual average and hourly time series of CO from BRAIN (a), MERRA-2 (b), and Sentinel-5P TROPOMI (daily averages) (c) in Porto Velho – Brazil.

To analyze BRAIN's performance in background regions (with low anthropogenic influence), we extracted data from two forested sites in the Brazilian North region. We used as reference the sampling sites of the GoAmazon experiment (Martin et al., 2016) named T0a (forested site situated 154.1 km from the Manaus urban area) and T0t/TT34 (a Broken canopy, forested site situated 60.9 km from the Manaus). Sentinel-5P TROPOMI data spatially aligned to BRAIN resolution was also extracted

for comparison. A buffer of 0.2° around the sites selected the data of CO, $O_3$, and $NO_2$ from both datasets. The results revealed that BRAIN captured the seasonal profile at T0a (Figure 11), showing a moderate correlation with tropospheric column measurements of Sentinel-5P TROPOMI, especially for CO and $O_3$.

BRAIN estimates are slightly higher than observed concentrations in background areas of CO, $O_3$, and $NO_2$ in TT34 (Figure 12) and T0a (Figure 11). While $O_3$ concentrations simulated by BRAIN range around 18 ppb (average in 2019) at the TT34 site, observed concentrations in 2013 (Artaxo et al., 2013) were around 8.5 ppb ± 1.9 ppb. In T0a, BRAIN simulated concentrations around 16 ppb, overestimating the observations (7 ppb ± 2 ppb during the wet season from March to April 2013-2020) (Nascimento et al., 2022). Concerning CO, the concentrations simulated by BRAIN are slightly lower, ranging around 73 ppb (average) at TT34 against 130 ppb observed during the GoAmazon experiment from 2010 to 2011 (Artaxo et al., 2013). We emphasize that the BRAIN and GoAmazon datasets are reported in different periods and, consequently, influenced by different emissions rates. For instance, fire emissions have changed significantly since 2011 in Amazon (Copernicus, 2022; Naus et al., 2022).

We also analyzed BRAIN results in the Manaus urban area. We adopted the sampling site of the GoAmazon experiment (Martin et al., 2016) named T1 (INPA campus in Manaus). Compared with Sentinel-5P TROPOMI data, BRAIN fairly reproduced the temporal pattern of CO, $O_3$, and $NO_2$ in the T1 site (Figure 13). Rafee et al., (2017) reported mean concentrations of 88.7 ppb of NOx and 382.6 pbb of CO in the Manaus urban area, while BRAIN reached 79 ppb and 99 ppb (maximum of 383 ppb), revealing an underestimation in this area. Again, the sampling campaign presented by Rafee et al. (2017) and BRAIN simulations uses different base year. Comparing BRAIN at T0a/TT34 (background sites) and T1 (urbanized), the database has reached consistent results with lower concentration levels in preserved areas.

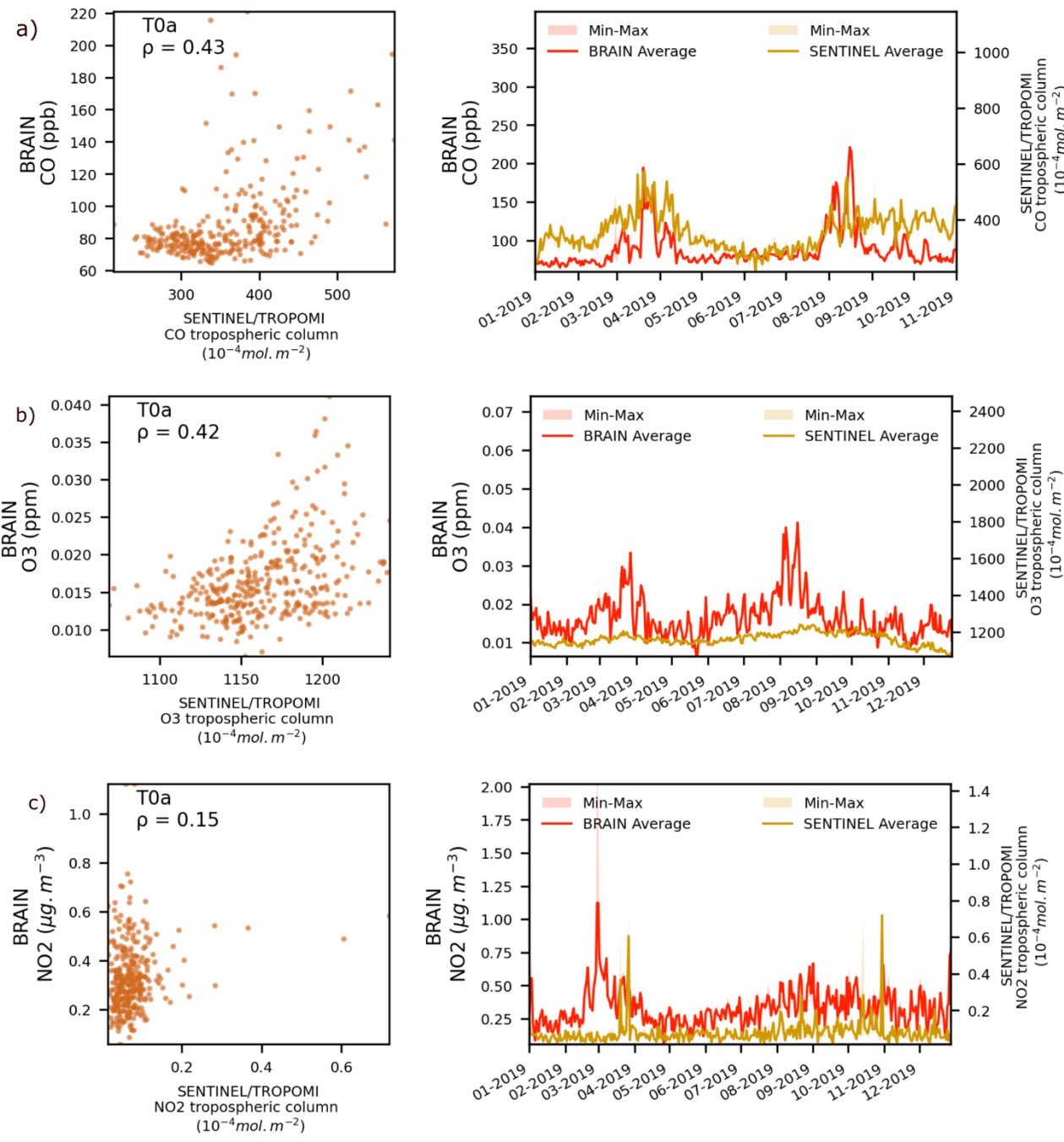

**Figure 11. Scatterplot and daily time series of CO (a), O₃ (b), and NO₂ (c) from BRAIN and Sentinel-5P TROPOMI at T0a (GoAmazon reference). Values extracted using a buffer of 0.2° around the site.**

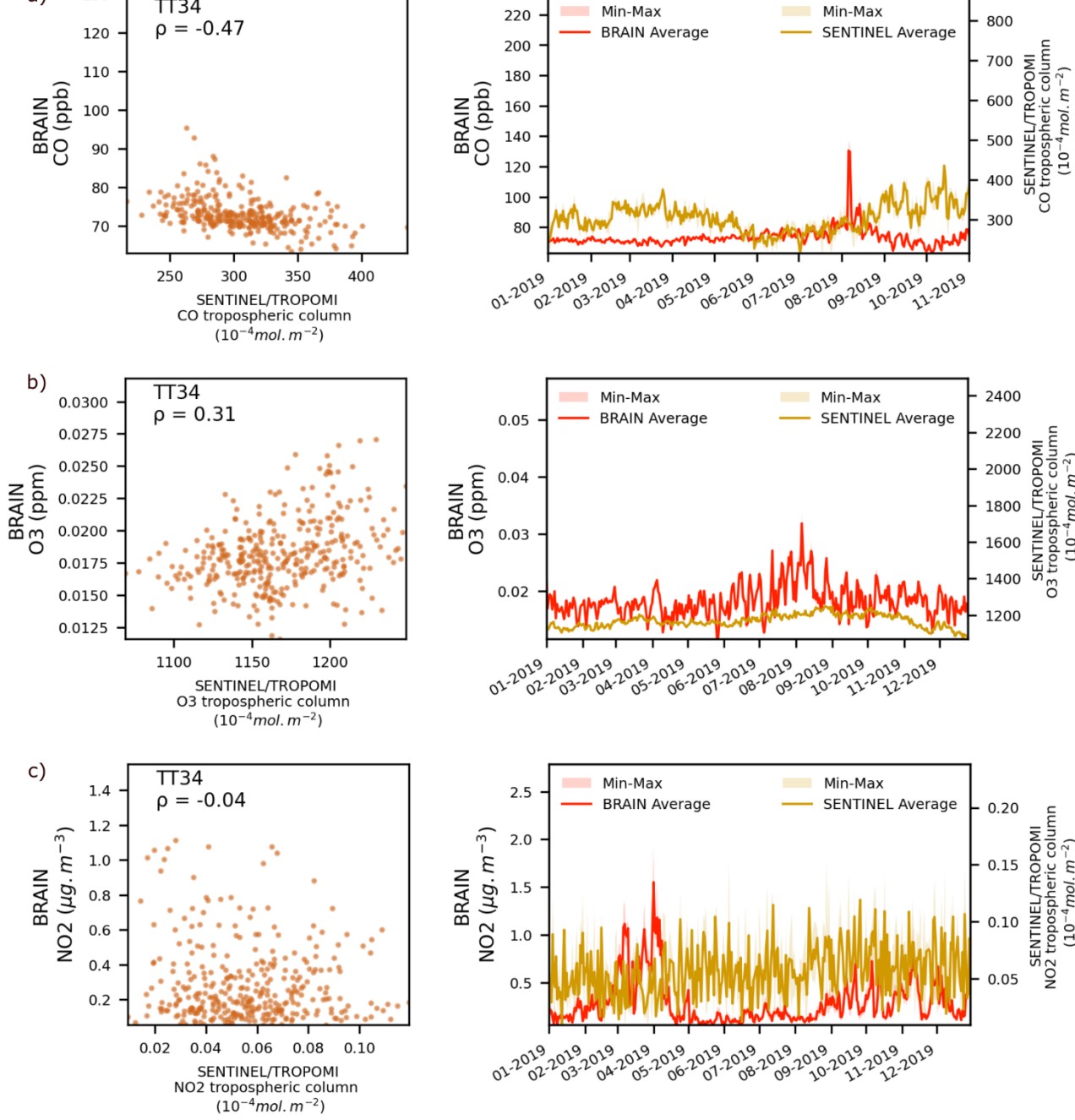

Figure 12. Scatterplot and daily time series of CO, O₃, and NO₂ from BRAIN and Sentinel-5P TROPOMI at T0t/TT34 (GoAmazon reference). Values extracted using a buffer of 0.2° around the site.

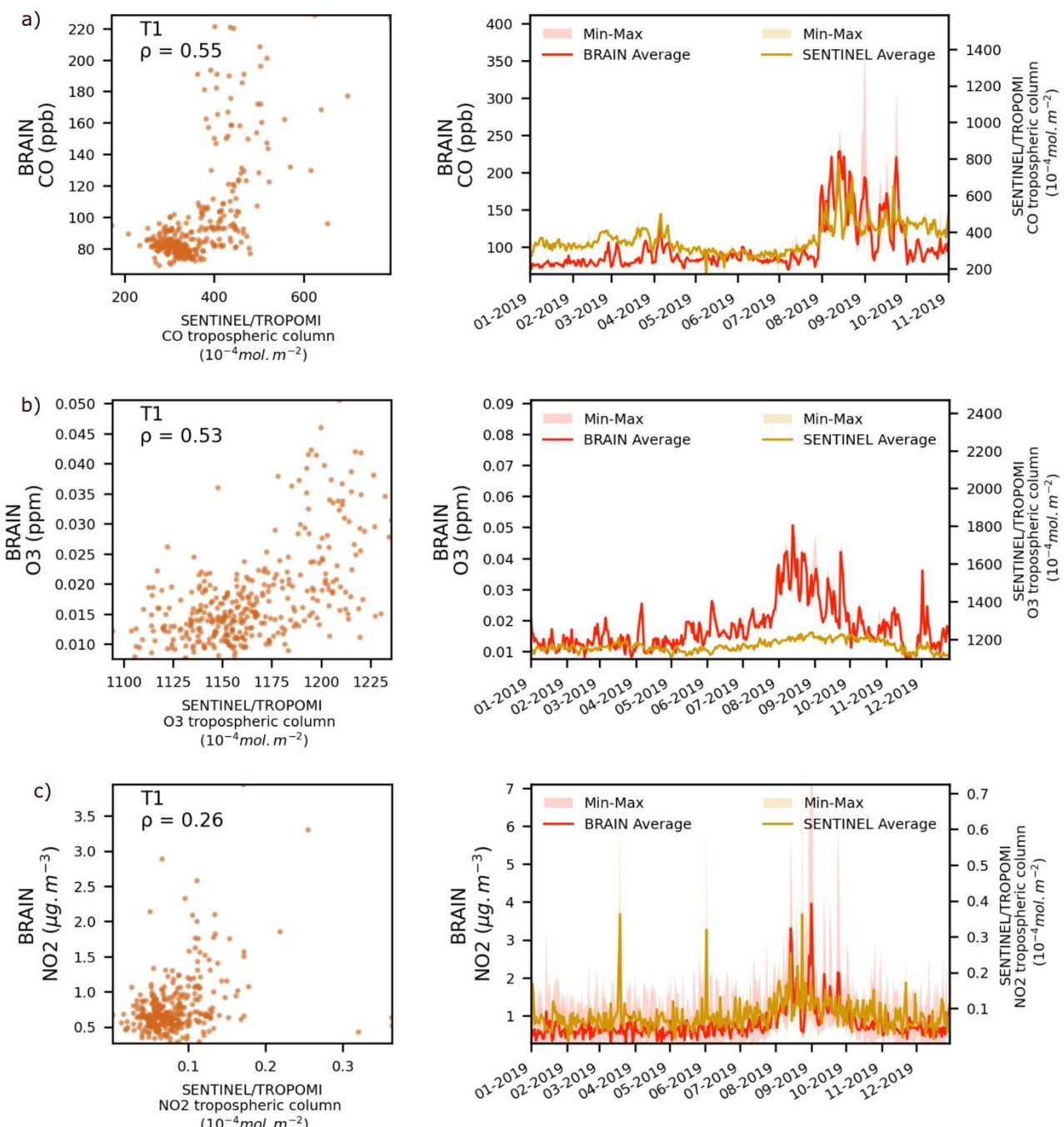

**Figure 13. Scatterplot and daily time series of CO (a), O₃ (b), and NO₂ (c) from BRAIN and Sentinel-5P TROPOMI at T1 (GoAmazon reference). Values extracted using a buffer of 0.2° around the site.**

The inability to better predict the observations relies mostly on the quality of the emissions inventory. The lack of information on industrial emissions and their temporal variability is an important source of errors. Moreover, the vehicular emissions inventory also needs improvements to properly disaggregate the emissions in high-flow roads. Future versions of BRAIN could address these issues and incorporate other emission sources.

## 3. Data availability

**Table 1.** BRAIN datasets freely available.

| Dataset | DOI | Reference | Citation |
|---|---|---|---|
| Emission d01 | 10.57760/sciencedb.09858 | Hoinaski, L., Will, R., Ribeiro, C.B. (2023a). Brazilian Atmospheric Inventories - BRAIN version 1: emission dataset in Brazil[DS/OL]. V1. Science Data Bank, 2023[2023-08-02]. https://cstr.cn/31253.11.sciencedb.09858. CSTR:31253.11.sciencedb.09858. | Hoinaski et al., (2023a) |
| Emission d02 | 10.57760/sciencedb.09886 | Hoinaski, L., Will, R., Ribeiro, C.B. (2023b). Brazilian Atmospheric Inventories - BRAIN version 1: emission dataset in Southern Brazil[DS/OL]. V1. Science Data Bank, 2023[2023-08-02]. https://cstr.cn/31253.11.sciencedb.09886. CSTR:31253.11.sciencedb.09886. | Hoinaski et al., (2023b) |
| Meteorology d01 | 10.57760/sciencedb.09857 | Hoinaski, L., Will, R. (2023a). Brazilian Atmospheric Inventories - BRAIN version 1: meteorology dataset in Brazil[DS/OL]. V1. Science Data Bank, 2023[2023-08-01]. https://cstr.cn/31253.11.sciencedb.09857. CSTR:31253.11.sciencedb.09857. | Hoinaski and Will, (2023a) |
| Meteorology d02 | 10.57760/sciencedb.09885 | Hoinaski, L., Will, R. (2023c). Brazilian Atmospheric Inventories - BRAIN version 1: meteorology dataset in Southern Brazil[DS/OL]. V1. Science Data Bank, 2023[2023-08-02]. https://cstr.cn/31253.11.sciencedb.09885. CSTR:31253.11.sciencedb.09885. | Hoinaski and Will, (2023c) |
| Air quality d01 | 10.57760/sciencedb.09859 | Hoinaski, L., Will, R. (2023b). Brazilian Atmospheric Inventories - BRAIN version 1: air quality dataset in Brazil[DS/OL]. V1. Science Data Bank, 2023[2023-08-01]. https://cstr.cn/31253.11.sciencedb.09859. CSTR:31253.11.sciencedb.09859. | Hoinaski and Will, (2023b) |
| Air quality d02 | 10.57760/sciencedb.09884 | Hoinaski, L., Will, R. (2023d). Brazilian Atmospheric Inventories - BRAIN version 1: air quality dataset in Southern Brazil[DS/OL]. V1. Science Data Bank, 2023[2023-08-02]. https://cstr.cn/31253.11.sciencedb.09884. CSTR:31253.11.sciencedb.09884. | Hoinaski and Will, (2023d) |

## 4. Code availability

Codes to generate the database, statistic, and figures are available at: https://github.com/leohoinaski/CMAQrunner (last access: 27 July 2023).

## 5. Conclusion

In this paper, we present BRAIN, the first comprehensive database for air quality management in Brazil. BRAIN provides emissions, meteorology, and air quality datasets for the entire country in reliable spatiotemporal resolution. BRAIN database covers a wide range of pollutant species (emissions and ambient concentrations) and atmospheric variables. So far, Brazil has lacked a comprehensive and easily accessible database for developing air quality management systems in urbanized and rural areas. This work contributes to overcoming this gap. BRAIN is a step forward for a good procedure for licensing new sources of air pollution in Brazil.

Using a sample of BRAIN, we observed several violations of WHO air quality recommendations. The violations are not restricted to densely populated areas but also occur in rural ones. It reinforces the need for better air quality policies and a deep restructuring of the environmental agencies' procedures and data management in Brazil.

Compared with observations, the BRAIN air quality dataset has achieved good overall performance in predicting the criteria pollutants. However, there is plenty of room for improvement mainly related to the quality of emissions inventory. The lack of information on industrial emissions and their temporal variability is an important source of errors. Moreover, the vehicular emissions inventory also needs improvements to properly disaggregate the emissions in high-flow roads. Improvements in boundary conditions and the inclusion of emissions sources from other Latin American countries could also enhance the CMAQ performance. The influence of long-range transport will be addressed in a future version of the database by implementing boundary contributions from GEOSCHEM and other tools. Future versions of BRAIN could address these issues, incorporate other emission sources, and provide CMAQ outputs using different chemical mechanisms. We envision providing enough data to reproduce the historical pattern and future scenarios of air pollution in Brazil through a web platform to facilitate the access and usage of our database. We believe in an ongoing process that will improve the database.

**Author contribution**

LH designed the methodology and developed the software. LH, RW, and CBR processed the data curation, formal analysis, and created the figures. LH, RW, and CBR prepared the original draft and revised the manuscript. LH is the project administrator and laboratory supervisor.

**Competing interests**

The authors declare that they have no conflict of interest.

**Acknowledgments**

The authors would like to thank the Secretaria de Estado do Desenvolvimento Econômico Sustentável do governo de Santa Catarina. The authors are grateful for the doctoral scholarships financed by the Coordenação de Aperfeiçoamento de Pessoal de Nível Superior – Brasil (CAPES) – Finance Code 001.

**Financial support**

This research has been supported by the Fundação de Amparo à Pesquisa e Inovação do Estado de Santa Catarina (grant no. 2018/TR/499) ("Avaliação do impacto das emissões veiculares, queimadas, industriais e naturais na qualidade do ar em Santa Catarina.").

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
