# Peer review of "Brazilian Atmospheric Inventories - BRAIN: A comprehensive database of air quality in Brazil"

_Earth System Science Data, 2023_

## Author Response (AR1)

**Brazilian Atmospheric Inventories - BRAIN: A comprehensive database of air quality in Brazil**

MS No.: essd-2023-305

Dear reviewers,

      We greatly appreciate the comments and suggestions, which are very constructive and have contributed to enhance the content of the revised manuscript.  We have uploaded a new version of BRAIN for 2020, updating versions of FINN and BRAVES. Datasets will be available at following links:

https://doi.org/10.57760/sciencedb.14569

https://doi.org/10.57760/sciencedb.14568

https://doi.org/10.57760/sciencedb.14560

https://doi.org/10.57760/sciencedb.14564

https://doi.org/10.57760/sciencedb.14563

https://doi.org/10.57760/sciencedb.14561

Please find below the reply to all the reviewer's comments.

Best regards

Leonardo Hoinaski and coauthors

**Reply to comments by Reviewer #1:**

Reply to general comments

General comment:

The manuscript shows the development and analysis of an air quality management system to help air quality policies and management in Brazil. The paper brings a very interesting topic, specfically due to: i) the current lack of numerical simulated dataset needed to work with air quality in Brazil; ii) Not everyone that works with meteorological and/or air quality in knows how to produce and work with atmospheric models. Thus, having the possibility to use a platform, as proposed by the authors for a specific region, is an indubitable advantage. I would recommend the paper for publishing after addressing all the following comments.

My major concern is on creating numerical simulation for a country with such a large size as Brazil. For that, a massive statistic evaluation is mandatory and not only in cities with large amount of pollutants emissions but also in background regions, such as the Northwest of the Amazonas state (near to the Amazon Tall Tower Observatory -ATTO). In cases where a monitoring network is not available, the authors should use remote sensing to evaluate the numerical experiments. There is a good amount of satellite products to use in this case: MODIS to evaluate AOD; TROPOMI to evaluate NO2 and HCHO; IASI+GOME2 to evaluate O3 (on the 20km grid). If the authors are proposing to make available numerical simulations for the entire country, it requires evaluation for the whole domain or at least an evaluation scale where it shows a confiability level of the data. In order to assume the premises of good modeling results for regions without validation, the authors need to build an underlying analysis by showing how the model performs on key scenarios, such as:

i) Background regions.

ii) Under anthropogenic emission effects (also analyze the model's abilities with plume transport).

ii) Long range transport (specially in the Amazon region during the wet season with (between January and May), when the Intertropical Convergence Zone (ITCZ) is more intense in the south, allowing the long transport of BC (biomass burning emissions) and dust from Africa (Sahara desert) (Artaxo et al., 2013; Martin et al., 2016; Pöhlker et al., 2018, 2019). With the availability of several years of BC background measurements at the ATTO tower, the authors could separate African episodic events from the rather constant regional BC concentrations that are relevant when comparing with modeled values not under anthropogenic influence.

Once it is shown that the model is capable of representing key scenarios, the idea of using modeling data for regions not fully evaluated is more reasonable.

Reply: We appreciate the positive feedback and all points raised by the reviewer, which will improve our manuscript and give new directions for the continuity of our work. The BRAIN air quality dataset evaluation uses observations extracted from 244 air quality monitoring stations in Brazil in nine Brazilian states. We agree that BRAIN must be tested in background regions and other unmonitored areas. A new comparison with gridded data from MERRA-2 and TROPOMI has been included in the final version of the manuscript, showing the model's performance in background regions. We will also demonstrate that BRAIN can represent key scenarios capturing observed temporal patterns in background regions, comparing with the references you listed. The influence of long-range transport will be addressed in a future version of the database by implementing boundary contributions from GEOSCHEM and other tools. We thank you for the list of references which improves the discussion in our manuscript.

We addressed the following points regarding BRAIN performance on key scenarios:

**i) Background regions**

Artaxo et al., (2013) presents an analysis of observed aerosols at a preserved forest site in Central Amazonia (TT34 North of Manaus) and at a heavily biomass burning impacted site in south-western Amazonia (PVH, close to Porto Velho) (https://pubs.rsc.org/en/content/articlepdf/2013/fd/c3fd00052d). We observed a similar seasonal pattern when comparing BRAIN with results from sampling campaign presented by Artaxo (2013) in Manaus and Porto Velho for $PM_{2.5}$, $O_3$, and CO.

Comparing Figure 3 from Artaxo et al., (2013) (top) and $PM_{2.5}$ time-series in Manaus (SM8) from BRAIN (bottom), we observe $PM_{2.5}$ concentrations (fine) in North Manaus in agreement with BRAIN in the same area showing similar absolute magnitude of BRAIN's average values. We can also observe that BRAIN captured the spatial variability in Manaus, detecting minimum concentrations in North where the forest is preserved. It shows de BRAIN can represent scenarios of background concentrations.

**Figure 3 from Artaxo et al., (2013). Time series of $PM_{2.5}$ in central Amazonia/North Manaus.**

[Figure]

**Fig. 3** Time series of fine ($PM_{2.5}$) and coarse mode aerosol mass concentrations at the central Amazonia TT34 forest site from 2008 to 2012.

**Figure in SM8 from the present manuscript. Average concentration and time series in Manaus. BRAIN captured smaller concentrations in North Manaus.**

[Figure]

We included the following sentence in the manuscript to highlight the similarities between BRAIN and the sampling campaign used in Artaxo et al. (2013):

> *BRAIN captures seasonal patterns and the absolute magnitude of PM$_{2.5}$ in the Northwest of the Amazonas state (near the Amazon Tall Tower Observatory -ATTO) presented by Artaxo et al. (2013). It shows that our database can reproduce the concentrations in background areas (far from highly urbanized centers).*

We prepared a new evaluation of BRAIN using MERRA-2 and Sentinel air quality datasets. We included the following sentences and figures in the manuscript:

> *BRAIN captures seasonal patterns and the absolute magnitude of PM2.5 in the Northwest of the Amazonas state (near the Amazon Tall Tower Observatory -ATTO) presented by Artaxo et al., (2013). It shows that our database can reproduce the concentrations in background areas (far from highly urbanized centers). Comparing BRAIN with observations at heavy biomass burning impacted sites in south-western Amazonia (Porto Velho) (Artaxo et al., 2013) revealed that BRAIN can capture seasonal variations caused by wet and dry seasons and the magnitude of average and peak concentrations.*
>
> *BRAIN has a similar spatial pattern compared with MERRA-2 (GMAO, 2015a,b), capturing hotspots in higher populated areas located in the Southeast, South, and Mid-West. In the Amazon region, BRAIN can also capture hotspots similar to MERRA-2 (Figure 6). BRAIN estimates for carbon monoxide are lower than MERRA-2, except for the South region and some urban centers in the Southeast and Midwest (Figure 6). Carbon monoxide concentrations estimated by BRAIN are moderately correlated with MERRA-2 mainly in the South (0.57) and Southeast (0.55), while in the Midwest, North, and Northeast the correlation is weaker (Figure 7). Compared with the consolidated MERRA-2 database, BRAIN has the advantage since it uses local and more refined information and provides data in higher spatial resolution for multiple species.*

[Figure]

*Figure 6. Annual average concentration of CO from BRAIN original resolution (a), BRAIN regridded to MERRA2 resolution (b), MERRA2 (c), and difference between MERRA2 and BRAIN (d).*

*We provide a detailed comparison between MERRA-2 and BRAIN datasets for PM$_{2.5}$, SO$_2$, O$_3$, and CO in SM16.*

[Figure]

*Figure 7. Concentration of CO from BRAIN vs MERRA2 in Brazil (a), North Brazil (b), Northeast Brazil (c), Midwest Brazil (d), Southeast Brazil (e), South Brazil (f).*

*We also compare our database with SENTINEL/TROPOMI (Veefkind et al., 2012) campaigns to demonstrate BRAIN's ability to capture the spatiotemporal variability of air pollutants in unmonitored areas (Figure 8). We regridded the SENTINEL/TROPOMI products to BRAIN resolution, using data downloaded from link. We merged all layers of the same day and interpolated to match BRAIN resolution. We computed the daily averages for both datasets. In this evaluation, we*

*must consider the differences between the datasets, since TROPOMI relies on tropospheric column measurements and BRAIN surface concentrations. The spatial analysis revealed that BRAIN captured the hotspots of CO and NO$_2$ in SENTINEL/TROPOMI products, especially in Southeast Brazil. However, the hotspots of CO are dislocated towards the ocean in SENTINEL/TROPOMI. $_{NO2}$ estimates from BRAIN present a higher number of hotspots. We emphasize that surface concentrations are better than tropospheric column in representing air pollution and its effects.*

[Figure]

*Figure 8. Annual average concentration of CO and NO$_2$ from BRAIN original resolution (a, c), SENTINEL/TROPOMI regridded to BRAIN resolution (b, d).*

*When we compared CO daily datasets from BRAIN and SENTINEL/TROPOMI by Brazilian regions, we observed a moderate correlation in North (0.41), Midwest (0.32), and South (0.3). This analysis shows that BRAIN can reasonably detect temporal and spatial patterns of air pollutants. The complete comparison of CO and NO$_2$ from SENTINEL/TROPOMI and BRAIN can be found in SM17.*

[Figure]

*Figure 9. Concentration of CO from BRAIN vs SENTINEL/TROPOMI in Brazil (a), North Brazil (b), Northeast Brazil (c), Midwest Brazil (d), Southeast Brazil (e), South Brazil (f).*

*We highlight that BRAIN, MERRA2, and SENTINEL/TROPOMI can capture similar temporal patterns of air pollution in heavy biomass-burning impacted sites such as Porto Velho in Rondônia (Figure 10), background areas such as Manaus in Amazonas, and urban areas such as São Paulo. We provide figures with the time series of BRAIN, MERRA2, and SENTINEL/TROPOMI of Brazilian capitals in SM18.*

[Figure]

*Figure 10. Annual average and hourly time series of CO from BRAIN (a), MERRA-2 (b), and Sentinel-5P TROPOMI (daily averages) (c) in Porto Velho – Brazil.*

**References**

Artaxo, P., Rizzo, L.V., Brito, J.F., Barbosa, H.M., Arana, A., Sena, E.T., Cirino, G.G., Bastos, W., Martin, S.T., Andreae, M.O., (2013). Atmospheric aerosols in Amazonia and land use change: from natural biogenic to biomass burning conditions. Faraday Discuss., 165, 203–235. https://doi.org/10.1039/C3FD00052D

ii)     **Under anthropogenic emission effects (also analyze the model's abilities with plume transport).**

Figure 5 from Artaxo (2013) brings PM time-series in Porto Velho. BRAIN time series of $PM_{2.5}$ in Porto Velho from 2019 can be found in supplementary material (SM8). Comparing both figures, we can observe a seasonal pattern with peaks in dry season (from August to October). It reveals that BRAIN captured seasonal variation and absolute magnitude of observations.

**Figure 5 from Artaxo et al. (2013). Time series of $PM_{2.5}$ in Porto Velho.**

[Figure]

Fig. 5   Time series of fine and coarse mode aerosol mass concentrations at the PVH anthropogenic impacted site, from 2009 to 2012.

**Figure in SM8 from the present manuscript. Average concentration and time series in Porto Velho. BRAIN captured seasonal pattern and the absolute magnitude of $PM_{2.5}$ in Porto Velho.**

[Figure]

We also observed similar magnitude in BRAIN and Artaxo (2013) for $O_3$ (Figure 14 from Artaxo's work) and CO (Figure 16 from Artaxo's work) in Porto Velho, as well as the seasonal variability.

**Figure 14 from Artaxo et al. (2013). Seasonal variability of O₃ in Porto Velho.**

[Figure]

**Fig. 14** Seasonal variability (2009–2012) of ozone volume mixing ratios in Porto Velho (PVH) and central Amazonia (Manaus TT34 tower). Circles represent median values, and bars represent 10 and 90 percentiles.

**Figure in SM8 from the present manuscript. Average concentration and time series of O₃ in Porto Velho. BRAIN captured seasonal pattern and the absolute magnitude of O₃ in Porto Velho.**

[Figure]

**Figure 14 from Artaxo et al. (2013). Seasonal variability of CO in Porto Velho.**

[Figure]

**Figure in SM8 from the present manuscript. Average concentration and time series of CO in Porto Velho. BRAIN captured seasonal pattern and the absolute magnitude of CO in Porto Velho.**

[Figure]

We included the following sentence in the manuscript to highlight the similarities between BRAIN and the sampling campaign used in Artaxo et al. (2013):

*Comparing BRAIN with observations at heavy biomass burning impacted sites in south-western Amazonia (Porto Velho) (Artaxo et al., 2013) revealed that BRAIN is capable of capturing seasonal variations caused by wet and dry seasons and the magnitude of average and peak concentrations.*

We also reinforced that BRAIN well reproduced the concentrations in moderate urbanized areas, such as Limeira and Piracicaba. The database reached moderate performance in highly urban areas such as Copacabana/RJ and at Marginal Tietê in the megacity of São Paulo. Please, see the figures below extracted from SM12. Regarding industrialized areas, BRAIN performed well in representing concentrations in the Pecém Industrial and Port Complex (CIPP). We included the following sentence in the manuscript:

*BRAIN well reproduced the concentrations in moderate urbanized areas, such as Limeira and Piracicaba (Figures in SM12). The database reached moderate performance in highly urban areas such as Copacabana/RJ and at Marginal Tietê in the megacity of São Paulo (Figure in SM12).*

*It reveals that the database can capture temporal patterns of air pollutants concentration in urbanized and industrialized areas.*

[Figure]

[Figure]

iii) Long range transport

We included the following sentences in the manuscript:

*The influence of long-range transport will be addressed in a future version of the database by implementing boundary contributions from GEOSCHEM and other tools.*

Reply to specific comments

I would strongly recommend the authors to put more focus on the states with the most extensive networks of AQS such as SP, RJ and MG, in addition of course to PR, SC and RS (I am suspicious it was done that way because the authors are based in SC).

Reply: We provide a detailed model evaluation by monitoring stations in nine Brazilian states, including SP, RJ, and MG. We used statistical metrics to measure the model's performance in each station. We also present visual statistical analyses through scatter and line plots by AQS in SM 12-14. We have planned to provide datasets with higher spatial resolution in Southeast Brazil in future versions of our database. New sentences will be included in the manuscript to clarify that we will provide further evaluation of BRAIN and more refined datasets for Southeast Brazil.

We included the following sentences in the manuscript:

*We envision making available more detailed datasets for other Brazilian regions, especially in the Southeast where the anthropogenic emission effects are more prominent. Future versions will also provide more detailed modeling outputs to properly cover medium- and small-sized cities.*

The authors should have a modeling strategy component focused on the States with AQS to allow the increase of the spatial resolution, and thus, get a more robust statistical evaluation for medium and small-sized cities that, otherwise, would not be properly covered by coarse resolution model simulations (10 km or more). I also recommend the authors to do a full evaluation by selecting a region with AQS available and combine observational data from satellites and field campaigns (if available). The idea here is to try to have a case where the model can be massive evaluated vertically and horizontally.

Reply:   We agree that the coarser domain does not properly cover small municipalities. It will be addressed in future versions by providing datasets with higher spatial resolutions.  We envision making available more detailed datasets for regions other than southern Brazil.  It is worth emphasizing that BRAIN will be a long-term project. These considerations will be included in the manuscript.
Besides the evaluation using 244 AQS, we prepared a spatial comparison of BRAIN with the MERRA-2 and TROPOMI datasets to evaluate the database in background areas.  A Full statistical evaluation with AQS has been presented in SM 11-14, showing that BRAIN performs adequately even with 20x20 km of resolution. It will be a good starting point for understanding the air pollution process in Brazil.

We included the following sentences in the manuscript:

*We envision making available more detailed datasets for other Brazilian regions, especially in the Southeast where the anthropogenic emission effects are more prominent. Future versions will also provide more detailed modeling outputs to properly cover medium- and small-sized cities.*

If there is no observations to compare with in States other than those with AQS, I do not think a country-scale simulation is really worthwhile here. Maybe the authors should provide a strategy/approach for validade/evaluate regions without AQS (remote sensing?).

Reply: For permitting and licensing purposes, the United States Environmental Protection Agency recommends the inclusion of background concentrations as part of a cumulative impact analysis. According to the Title 40 of the USA Code of Federal Regulations (appendix-W part 51), "*in those cases where adequately representative monitoring data to characterize background concentrations are not available, it may be appropriate to use results from a regional-scale photochemical grid model, or other representative model application*". In this context, BRAIN is truly worthwhile and could be a step forward for a good procedure for licensing new sources of air pollution. We understand that BRAIN requires a deeper evaluation. We believe in an ongoing process that will improve the database.  We emphasize that BRAIN has limitations and requires further evaluation.

We prepared a spatial comparison of BRAIN with the MERRA-2 and TROPOMI datasets to evaluate the database in unmonitored areas. It will be included in the manuscript as well as the considerations about the purpose of the database and its limitations.

We included the following sentences in the manuscript:

*BRAIN intends to fill the gaps in those cases where adequately representative monitoring data to characterize the air quality is not available. BRAIN would be useful to provide background concentrations for a good procedure for licensing new sources of air pollution.*

*BRAIN is a step forward for a good procedure for licensing new sources of air pollution in Brazil.*

*We believe in an ongoing process that will improve the database.*

As most of the air quality stations used for model evaluation are placed in southeastern Brazil, why not focusing on the largest metropolitan areas of the country, such as, the metropolitan areas of São Paulo, Rio de janeiro, Belo horizonte, etc?. Or maybe use passed campaigns such as The Green Ocean Amazon experiment (GoAmazon2014/5) or The Regional Carbon Balance in Amazonia (BARCA) to evaluate the chemistry component of the model in different vertical levels and with high spatial resolution.

Reply: Our goal with BRAIN is to provide air quality and emission datasets to overcome data gaps and the lack of spatial representativeness of AQS. We aim to fulfill the requirements of dose-response studies and permitting procedures for new sources in Brazilian municipalities. There is extensive literature addressing the air quality in great metropolitan areas. We have decided to use a wider approach to somehow contribute to the literature. Future analysis with BRAIN could evaluate the chemistry and physics of air pollution in targeted areas. We will clarify the focus/goals of our database in the manuscript. Also, we will include considerations about BRAIN's limitations and future perspectives.

We included the following sentences in the manuscript:

*We envision making available more detailed datasets for other Brazilian regions, especially in the Southeast where the anthropogenic emission effects are more prominent. Future versions will also provide more detailed modeling outputs to properly cover medium- and small-sized cities.*

*There is plenty of room for improvement mainly related to the quality of emissions inventory. The lack of information on industrial emissions and their temporal variability is an important source of errors. Moreover, the vehicular emissions inventory also needs improvements to properly disaggregate the emissions in high-flow roads. Improvements in boundary conditions and the inclusion of emissions sources from other Latin American countries could also enhance the CMAQ performance. Future versions of BRAIN could address these issues, incorporate other emission sources, and provide CMAQ outputs using different chemical mechanisms.*

*BRAIN intends to fill the gaps in those cases where adequately representative monitoring data to characterize the air quality is not available. BRAIN would be useful to provide background concentrations for a good procedure for licensing new sources of air pollution.*

The authors claimed that, currently available initiatives including reanalysis and satellite products are still not providing datasets with large spatial and temporal resolutions for developing air pollution policies in Brazil. In the biomass emission perspective, have the authors checked the fire products from Visible Infrared Imaging Radiometer Suite (VIIRS) on board the Suomi NPP satellite - VIIRS (375 m resolution for fire activities). The paper Ferrada et al., 2022 (Introducing the VIIRS-based Fire Emission

Inventory version 0 (VFEIv0)) shows a new open biomass burning inventory that relies on the fire radiative power (FRP) data from VIIRS.

Reply: We appreciate the recommendation. Indeed, VIIRS brings more detailed information and could potentially reproduce estimations with smaller uncertainties. We have prepared BRAIN since 2020. All codes/scripts were developed for data available in 2020. We have implemented the outdated FINNv1.5 in this first version. However, new data sources could be included in our emissions inventory in future work, which will optimize the database. We just finished a new version of BRAIN for 2020 using the new version of FINN and BRAVES. Therefore, the issue with outdated FINN has been already addressed. We will clarify that the database has the potential to incorporate new sources of data in future versions.

We included the following sentences in the manuscript:

> *We have implemented the FINNv1.5 in this first version of BRAIN. However, FINN version 2.5 will be included in our emissions inventory in future work, which uses an updated algorithm for determining fire size based on MODIS and VIIRS satellite instruments. We also provide data from 2020 with the same modeling grid upgrading to FINN v2.5.*

BRAVES uses activity data from field campaigns conducted in the metropolitan area of São Paulo. It really does not make any sense take this region out of a very high-resolution simulation design.

Reply: BRAVES has been developed by our group since 2018 to provide multispecies and high-spatiotemporal-resolution data of vehicular emissions covering the entire country. BRAVES totally suits this work. VEIN model (https://gmd.copernicus.org/articles/11/2209/2018/) is the one that uses activity data from field campaigns conducted in the metropolitan area of São Paulo. We will include more detailed information of activity data used in BRAVES modelling.

We included the following sentences in the manuscript:

> *In BRAVES, vehicular activity is defined by fuel consumption in each municipality using data provided by the Brazilian National Agency for Oil, Natural Gas and Biofuel (ANP) (https://www.gov.br/anp/pt-br/centrais-de-conteudo/dados-abertos/vendas-de-derivados-de-petroleo-e-biocombustiveis). A fraction of fuel consumed by road transportation is based on data from National Energy Balance (BEN) (https://www.epe.gov.br/pt/publicacoes-dados-abertos/publicacoes/balanco-energetico-nacional-ben) and (MMA, 2014). BRAVES calculates weighted EF to address the effect of the fleet composition in terms of category, model-year, and fuel utilization.*

NCAR (FINN) version 1.5 (Wiedinmyer et al., 2011). This version is pretty outdated. An updated version is now available https://egusphere.copernicus.org/preprints/2023/egusphere-2023-124/egusphere-2023-124.pdf.

Reply: Indeed, FINNv1.5 is pretty much outdated. We just finished a new version of BRAIN for 2020 using the new version of FINN fixing this issue. We plan to compare different sources of emission data in future versions of our database. We appreciate the recommendation, and we believe that it will strengthen BRAIN.

We included the following sentences in the manuscript:

> *We have implemented the FINNv1.5 in this first version of BRAIN. However, FINN version 2.5 will be included in our emissions inventory in future work, which uses an updated algorithm for determining fire size based on MODIS and VIIRS satellite instruments. We also provide data from 2020 with the same modeling grid upgrading to FINN v2.5.*

The authors mentioned that he WRF model demonstrated the ability to reproduce diurnal and seasonal variability of winds in the Brazilian North-East region (Souza et al., 2022a). Although, this resolution is slightly lower than the one used in this work for the parent domain. If wouldn't be better to just focus on the largest metropolitan areas of Brazil?. I am assuming that you have set up the model simulations at 20 km resolution in an attempt to avoid out-of-memory and space of storage issues, but if that is the case, why not just focusing on high densely areas?

Reply: There is a huge gap in air quality data (surface/ground level concentration) in Brazil, where most municipalities could not derive their own air quality management system, which motivated the BRAIN development. We focused on providing data on a national scale as a starting point for licensing new sources. As we did for Southern Brazil, we will reproduce datasets with better resolution for other Brazilian regions. We do have storage and processing limitations. Even though we have struggled with our small budget and team, we believe that BRAIN contributes to our field of knowledge. We will reinforce the BRAIN focus and where it could potentially be used.

We included the following sentences in the manuscript:

> *We envision making available more detailed datasets for other Brazilian regions, especially in the Southeast where the anthropogenic emission effects are more prominent. Future versions will also provide more detailed modeling outputs to properly cover medium- and small-sized cities.*
>
> *BRAIN intends to fill the gaps in those cases where adequately representative monitoring data to characterize the air quality is not available. BRAIN would be useful to provide background concentrations for a good procedure for licensing new sources of air pollution.*

The authors mentioned that the lack of data quality assurance may compromise the credibility of the available air quality observations in Brazil. It is true, and consequently, this could potentially compromise any analysis conducted on these data sets. In my view, a modeling study that centers on the States with the most extensive networks of AQS would have had a more effective simulation strategy, as the one mentioned in the previous comments. This point also brings the importance of using data from previous campaigns with high quality assurance.

Reply: Indeed, modeling study that centers on the States with the most extensive networks of AQS can be better evaluated. However, we have focused on providing data where AQS is not available. We believe that our database brings essential information for licensing new sources and deriving air quality management systems in Brazil. Further evaluation will bring robustness to our database. We will clarify the future perspectives of BRAIN in the manuscript.

We included the following sentences in the manuscript:

*BRAIN intends to fill the gaps in those cases where adequately representative monitoring data to characterize the air quality is not available. BRAIN would be useful to provide background concentrations for a good procedure for licensing new sources of air pollution.*

**Reply to comments by Reviewer #2:**

Reply to general comments.

The current paper aims to develop an emission database for the entire country of Brazil based on local emissions inventories for industries, vehicular emissions estimations, MEGAN for biogenic emissions, and FINN for fire emissions. Additionally, the authors aim to make this dataset available to the community.

The initiative is really good not only for Brazil but also for South America. The countries inside South America have their connection and impact of long-range transport of pollutants. Thus, an emissions inventory for the largest country is appreciated. Primarily when the government and environmental protection agencies do not provide the official data for the community

The authors have put much effort into the work, gathering all data (meteorology, emissions, temporal profile, chemical profile, different models, different databases), processing the models, and performing the air quality modeling.

The database is interesting and valuable for Brazil and South America and will be very appreciated by the community. As mentioned by the authors, the lack of this information holds the improvement in the air quality across Brazil. I would recommend the paper for publishing after addressing the following comments.

Reply:   We deeply appreciate the positive feedback.

Reply to specific comments

**1. Introduction**

The introduction shows the situation of the region of study regarding air quality problems, fire emissions, countryside problems with air pollution, and vehicle emissions across the country, among others. Additionally, the authors exposed the lack of air quality monitoring stations and emissions inventory. This situation is a concern for any country since the lack of this information is critical for air quality management.

The authors stated: "Currently, available initiatives, including reanalysis and satellite products, are still not providing datasets with large spatial and temporal resolutions for developing air pollution policies in Brazil." I would like some publications to enforce this statement. There are ERA reanalysis data with one hour of time resolution and less than 10km of spatial resolution. Additionally, there are MODIS data for air quality, probably not hourly, but there is a frequency that may help.

Reply:   We will include these initiatives in the introduction chapter, contrasting with BRAIN features.

We included the following sentences in the manuscript:

*Global reanalysis such as Copernicus Atmospheric Monitoring Service (CAMS) (Inness et al., 2018) and the second version of Modern-Era Retrospective analysis for Research and Applications (MERRA-2) (GMAO, 2015ab) can provide estimates of air pollutants by combining chemical transport models (CTMs) with satellite and ground-based observations and physical information, assimilating data to constrain the results. However, the currently available reanalysis products do not provide data with high spatial resolution (0.75° × 0.75° and 0.5° x 0.625°) and could be biased to represent local and regional air quality (Arfan Ali et al., 2022). Moreover, they provide data only for a small list of air pollutants. Satellite-based products such as Sentinel-5P TROPOMI (Veefkind et al., 2012) and Moderate-Resolution Imaging Spectroradiometer (MODIS) aboard Terra and Aqua satellites (Levy et al., 2015; Platnick et al., 2015) are still challenging due to their low temporal*

*resolutions, data gaps due to cloud coverage, and uncertainties (Shin et al., 2019). Besides, satellite data provide total column concentrations which do not represent surface concentrations (Shin et al., 2019).*

**References**

Arfan Ali, Md., Bilal, M., Wang, Y., Nichol, J.E., Mhawish, A., Qiu, Z., de Leeuw, G., Zhang, Y., Zhan, Y., Liao, K., Almazroui, M., Dambul, R., Shahid, S., Islam, M.N., (2022). Accuracy assessment of CAMS and MERRA-2 reanalysis PM2.5 and PM10 concentrations over China. Atmospheric Environment, 288, 119297. https://doi.org/10.1016/j.atmosenv.2022.119297

Global Modeling and Assimilation Office (GMAO) (2015a). MERRA-2 tavg1_2d_chm_Nx: 2d, 1-Hourly, Time-Averaged, Single-Level, Assimilation, Carbon Monoxide and Ozone Diagnostics V5.12.4, Greenbelt, MD, USA, Goddard Earth Sciences Data and Information Services Center (GES DISC). Accessed 01/08/2024. https://doi.org/10.5067/3RQ5YS674DGQ

Global Modeling and Assimilation Office (GMAO) (2015b). MERRA-2 tavg1_2d_aer_Nx: 2d, 1-Hourly, Time-averaged, Single-Level, Assimilation, Aerosol Diagnostics V5.12.4, Greenbelt, MD, USA, Goddard Earth Sciences Data and Information Services Center (GES DISC), Accessed 01/08/2024.  https://doi.org/10.5067/KLICLTZ8EM9D

Inness, A., Ades, Agusti-Panareda, A., Barré, J., Benedictow, A., Blechschmidt, A.M. et al., (2018). The CAMS reanalysis of atmospheric composition. Atmos. Chem. Phys. Discuss. 1-55. https://doi.org/10.5194/acp-2018-1078

Levy R., Hsu, C., et al., (2015). MODIS Atmosphere L2 Aerosol Product. NASA MODIS Adaptive Processing System, Goddard Space Flight Center, USA.

Platnick, S., Hubanks, P., Meyer, K., King, M.D., (2015). MODIS Atmosphere L3 Monthly Product (08_L3). NASA MODIS Adaptive Processing System, Goddard Space Flight Center.

Shin, M., Kang, Y., Park, S., Im, J., Yoo, C., Quackenbush, L.J., (2020). Estimating ground-level particulate matter concentrations using satellite-based data: a review. GIScience & Remote Sensing, 57:2, 174-189. https://doi.org/10.1080/15481603.2019.1703288

Veefkind, J.P., Aben, I., McMullan, K., Förster, H., de Vries, J., Otter, G., Claas, J., Eskes, H.J., de Haan, J.F., Kleipool, Q., van Weele, M., Hasekamp, O., Hoogeveen, R., Landgraf, J., Snel, R., Tol, P., Ingmann, P., Voors, R., Kruizinga, B., Vink, R., Visser, H., Levelt, P.F., (2012). TROPOMI on the ESA Sentinel-5 Precursor: A GMES mission for global observations of the atmospheric composition for climate, air quality and ozone layer applications. Remote Sensing of Environment, 120, 70-83. https://doi.org/10.1016/j.rse.2011.09.027

**2. BRAIN Database**

Error in the Figure 1 b): "Annual Indutrial"

Reply:   OK! We fixed this issue

2.1.1 Vehicular emissions

The BRAVES model was used to estimate the vehicle emissions. The database was validated in another paper, but I would like to see some explanation on how the authors estimated the activity data, especially over countryside cities. This information is crucial for understanding vehicular emissions.

Reply: BRAVES uses fuel consumption by the municipality to represent the activity data. We will include a more detailed description of BRAVES in the manuscript, explaining how the activity data has been estimated.

We included the following sentences in the manuscript:

> *In BRAVES, vehicular activity is defined by fuel consumption in each municipality using data provided by the Brazilian National Agency for Oil, Natural Gas and Biofuel (ANP) (https://www.gov.br/anp/pt-br/centrais-de-conteudo/dados-abertos/vendas-de-derivados-de-petroleo-e-biocombustiveis). A fraction of fuel consumed by road transportation is based on data from National Energy Balance (BEN) (https://www.epe.gov.br/pt/publicacoes-dados-abertos/publicacoes/balanco-energetico-nacional-ben) and (MMA, 2014). BRAVES calculates weighted EF to address the effect of the fleet composition in terms of category, model-year, and fuel utilization.*

***References***

ANP - Agência Nacional do Petróleo, Gás Natural e Biocombustíveis. Vendas de derivados de petróleo e biocombustíveis. Accessed 01/08/2024. Available at https://www.gov.br/anp/pt-br/centrais-de-conteudo/dados-abertos/vendas-de-derivados-de-petroleo-e-biocombustiveis

EPE - Empresa de Pesquisa Energética. Balanço Energético Nacional. Accessed 01/08/2024. Available at https://www.epe.gov.br/pt/publicacoes-dados-abertos/publicacoes/balanco-energetico-nacional-ben

MMA - Ministério do Meio Ambiente (2014). Inventário Nacional de Emissões Atmosféricas por Veículos Automotores Rodoviários 2013. Accessed 01/08/2024. Available at https://antigo.mma.gov.br/images/arquivo/80060/Inventario_de_Emissoes_por_Veiculos_Rodoviarios_2013.pdf

2.1.2 Industrial emissions

It used the emissions inventory of Espírito Santo, Minas Gerais, and Santa Catarina, which are local emissions inventory provided by the Environmental Protection Agency. Additionally, the emissions inventory developed by Kawashima et al. (2020) was also used. These emissions inventories are not from the same year and do not cover the entire state. Did the authors scale the emissions for the current modeling year, which is 2019? Please explain this in the text.

Additionally, have the authors considered applying EDGAR emissions across the regions without emissions inventory, especially for the industry sector?

Reply: We will clarify this point by including more details of the industrial inventory. We did not convert the emissions inventory to the current modeling year. We agree that it is a drawback in our database that must be addressed in further detail. Future efforts will be expended to improve the industrial inventory, scaling to the current year and including not inventoried sources.

We included the following sentences in the manuscript:

*We did not convert the emissions inventory to the current modeling year since the data is not continuously updated. Therefore, we assumed that all emissions from these multiple sources occurred in 2019.*

2.1.3 Biomass burning emissions

FINN is a good database for biomass burning, primarily for providing daily Biomass burning emissions. I noticed the authors used the FINN 1.5, but the FINN version 2.5 is available. I recommend the authors to use the updated version.

Reply: Indeed, FINNv1.5 is outdated. This issue has been fixed in a new version of BRAIN which is already available.

We included the following sentences in the manuscript:

*We have implemented the FINNv1.5 in this first version of BRAIN. However, FINN version 2.5 will be included in our emissions inventory in future work, which uses an updated algorithm for determining fire size based on MODIS and VIIRS satellite instruments. We also provide data from 2020 with the same modeling grid upgrading to FINN v2.5.*

2.1.5 Sea spray aerosol emissions

The Sea spray aerosol is handled inline in CMAQ. Is there a way to extract these values to use in another photochemical model, such as WRF-CHEM or CAMx?

Reply: We are not sure if one can extract sea spray emissions from CMAQ. We provide the OCEAN file (https://github.com/USEPA/CMAQ/blob/main/DOCS/Users_Guide/Tutorials/CMAQ_UG_tutorial_oceanfile.md) in our database. You can derive the sea spray emissions using this file. You can also derive your own OCEAN file using this Python script: https://github.com/leohoinaski/CMAQrunner/blob/master/hoinaskiSURFZONEv2.py

2.3 Air Quality & 2.3.1 Models' performance

The air quality was developed using CMAQ v5.3.2. The boundary conditions for the Brazilian domain are based on the standard profile. I would recommend that the authors provide Boundary conditions from a global model, such as GEOS-Chem, as mentioned in the paper, in this version or future work. This will account for the long-range transport of air pollutants across the country.

Regarding the model performance. The lack of air quality monitoring stations across Brazil was mentioned. However, there are some important areas in which the model should be validated for at least a few days across the modeling scenarios. This helps to understand if the emissions inventory should increase or decrease the total emissions. A validation with satellite data will improve this analysis.

Additionally, there is only the validation and plot for the 20x20km. How did the 4x4km modeling domain perform? I recommend adding this information to the text and supplementary material.

Reply: We agree that we should have included boundary conditions from a global model, such as GEOS-Chem. It will be completely addressed in future versions of our database. We have included a spatial comparison between the BRAIN dataset and MERRA-2 + TROPOMI to evaluate the database in unmonitored areas.

We included the following sentences in the manuscript:

*BRAIN captures seasonal patterns and the absolute magnitude of PM2.5 in the Northwest of the Amazonas state (near the Amazon Tall Tower Observatory -ATTO) presented by Artaxo et al. (2013). It shows that our database can reproduce the concentrations in background areas (far from highly urbanized centers). Comparing BRAIN with observations at heavy biomass burning impacted sites in south-western Amazonia (Porto Velho) (Artaxo et al., 2013) revealed that BRAIN is capable of capturing seasonal variations caused by wet and dry seasons and the magnitude of average and peak concentrations.*

*BRAIN has a similar spatial pattern compared with MERRA-2 (GMAO, 2015a b), capturing hotspots in higher populated areas located in the Southeast, South, and Mid-West. In the Amazon region, BRAIN can also capture hotspots similar to MERRA-2 (Figure 6). BRAIN estimates for carbon monoxide are lower than MERRA-2, except for the South region and some urban centers in the Southeast and Midwest (Figure 6). Carbon monoxide concentrations estimated by BRAIN are moderately correlated with MERRA-2 mainly in the South (0.57) and Southeast (0.55), while in the Midwest, North, and Northeast the correlation is weaker (Figure 7). Compared with the consolidated MERRA-2 database, BRAIN has the advantage since it uses local and more refined information and provides data in higher spatial resolution for multiple species.*

[Figure]

*Figure 6. Annual average concentration of CO from BRAIN original resolution (a), BRAIN regridded to MERRA2 resolution (b), MERRA2 (c), and difference between MERRA2 and BRAIN (d).*

*We provide a detailed comparison between MERRA2 and BRAIN datasets for PM$_{2.5}$, SO$_2$, O$_3$, and CO in SM16.*

[Figure]

*Figure 7. Concentration of CO from BRAIN vs MERRA2 in Brazil (a), North Brazil (b), Northeast Brazil (c), Midwest Brazil (d), Southeast Brazil (e), South Brazil (f).*

*We also compare our database with SENTINEL/TROPOMI (Veefkind et al., 2012) campaigns to demonstrate BRAIN's ability to capture the spatiotemporal variability of air pollutants in unmonitored areas (Figure 8). We regridded the*

*SENTINEL/TROPOMI products to BRAIN resolution, using data downloaded from link. We merged all layers of the same day and interpolated to match BRAIN resolution. We computed the daily averages for both datasets. In this evaluation, we must consider the differences between the datasets, since TROPOMI relies on tropospheric column measurements and BRAIN surface concentrations. The spatial analysis revealed that BRAIN captured the hotspots of CO and $NO_2$ in SENTINEL/TROPOMI products, especially in Southeast Brazil. However, the hotspots of CO are dislocated towards the ocean in SENTINEL/TROPOMI. $_{NO2}$ estimates from BRAIN present a higher number of hotspots. We emphasize that surface concentrations are better than tropospheric column in representing air pollution and its effects.*

[Figure]

*Figure 8. Annual average concentration of CO and $NO_2$ from BRAIN original resolution (a, c), SENTINEL/TROPOMI regridded to BRAIN resolution (b, d).*

*When we compared CO daily datasets from BRAIN and SENTINEL/TROPOMI by Brazilian regions, we observed a moderate correlation in North (0.41), Midwest (0.32), and South (0.3). This analysis shows that BRAIN can reasonably detect temporal and spatial patterns of air pollutants. The complete comparison of CO and $NO_2$ from SENTINEL/TROPOMI and BRAIN can be found in SMxxx e SM17.*

[Figure]

*Figure 9. Concentration of CO from BRAIN vs SENTINEL/TROPOMI in Brazil (a), North Brazil (b), Northeast Brazil (c), Midwest Brazil (d), Southeast Brazil (e), South Brazil (f).*

*We highlight that BRAIN, MERRA2, and SENTINEL/TROPOMI can capture similar temporal patterns of air pollution in heavy biomass-burning impacted sites such as Porto Velho in Rondônia (Figure 10), background areas such as Manaus in*

*Amazonas, and urban areas such as São Paulo. We provide figures with the time series of BRAIN, MERRA2, and SENTINEL/TROPOMI of Brazilian capitals in SM18.*

[Figure]

*Figure 10. Annual average and hourly time series of CO from BRAIN (a), MERRA-2 (b), and Sentinel-5P TROPOMI (daily averages) (c) in Porto Velho – Brazil.*

We provide a modeling evaluation for the 4x4km domain in the revised manuscript and included the following sentence in the manuscript:

*We analyzed the performances of 4x4 km simulations for CO, $NO_2$, $O_3$, and $SO_2$ drawing a buffer of 0.5° degrees around monitoring station positions in southern Brazil. Our findings indicated higher Spearman values for the spatial resolution of 20x20 km for CO, $O_3$, and $SO_2$. Specifically, for $O_3$, the best result at 20x20 km was 0.76, whereas the same point at 4x4 km resolution showed a correlation of 0.46. This pattern was also observed for CO, with the best result at 20x20 km being 0.47 for Spearman and 0.23 at the same point at 4x4 km resolution. The smallest differences in Spearman rank were observed for $SO_2$ (0.22: 20x20, 0.19: 4x4). Even though improving spatial resolution did not increase the correlation with measured data, we found best results for Bias, RMSE, and MAE for almost all pollutants at a 4x4 km resolution, except for CO. Please refer to SM15 for the complete statistical analysis of 4x4 km simulations.*

---

## Author Response (AR2)

**Brazilian Atmospheric Inventories - BRAIN: A comprehensive database of air quality in Brazil**

MS No.: essd-2023-305

Dear editor,

We greatly appreciate the reviewer comments and suggestions, which are very constructive and have contributed to enhance the content of the revised manuscript. Please find below the reply to all the reviewer comments.

Best regards

Leonardo Hoinaski and coauthors

**Reply to comments of reviewer:**

**Comment #1:** First of all, it is important for the authors to clarify the site locations. When

comparing the BRAIN with Artaxo et al., 2013 results.

Reply: We included new analyses comparing data from BRAIN and Sentinel-5P TROPOMI at the sites described in Martin et al., (2016) (background and urban areas). We adopted reanalysis and satellite data to compare the data from the same year as the BRAIN simulations, since GoAmazon campaigns are not available and Artaxo's work uses data from 2013.

We compared BRAIN and Sentinel-5P TROPOMI at T0t/TT34 (located at the ZF2 ecological reserve - low anthropogenic influence), T1 (Manaus - urban area), and T0a (upwind of Manaus – background area) sites as recommended by the reviewer. We used data from CO, $NO_2$, $O_3$ and $PM_{10}$ from BRAIN and tropospheric column of CO, $NO_2$, and $O_3$ from Sentinel-5P TROPOMI.

We have included the following sentences in the manuscript:

*To analyze BRAIN's performance in background regions (with low anthropogenic influence), we extracted data from two forested sites in the Brazilian North region. We used as reference the sampling sites of the GoAmazon experiment (Martin et al., 2016) named T0a (forested site situated 154.1 km from the Manaus urban area) and T0t/TT34 (a Broken canopy, forested site situated 60.9 km from the Manaus). Sentinel-5P TROPOMI data spatially aligned to BRAIN resolution was also extracted for comparison. A buffer of 0.2° around the sites selected the data of CO, $O_3$, and $NO_2$ from both datasets. The results revealed that BRAIN captured the seasonal profile at T0a (Figure 11), showing a moderate correlation with tropospheric column measurements of Sentinel-5P TROPOMI, especially for CO and $O_3$. BRAIN estimates are consistent with concentrations in background areas of CO (up to ~ 350 ppb), $O_3$ (up to ~ 0.04 ppm), and $NO_2$ (up to ~ 2.5 $\mu g.m^{-3}$) at T0t/TT34 (Figure 12) and T0a. However, it was not able to capture the seasonal profile at T0t/TT34.*

*We also analyzed BRAIN results in the Manaus urban area. We adopted the sampling site of the GoAmazon experiment (Martin et al., 2016) named T1 (INPA campus in Manaus). Compared with Sentinel-5P TROPOMI data, BRAIN fairly reproduced the temporal pattern of CO, $O_3$, and $NO_2$ in the T1 site (Figure 13).*

**Comment #2:** Manaus is not a background region. During the GoAmazon 204/5 experiment (Martin et al., 2016) many different sites were used to understand how the interaction between biogenic and urban emissions can affect the aerosol life cycle and its impacts on the cloud, radiation and water cycle in the Amazon region. To investigate background regions during the GoAmazon experiment the authors should look at sites: T0a (ATTO), T0z (TT34-ZF2), T0e. Those sites were upwind of Manaus, which means almost no influence from Manaus anthropogenic emission. T1 site (located in Manaus) is the site that was representing the gas/particle distribution in the city.

Reply: Thanks for the suggestions. We compared data of BRAIN and Sentinel-5P TROPOMI during 2019 at the different sites as suggested. To analyze BRAIN results in background regions, we adopted a forested site upwind of Manaus (named T0a in the GoAmazon reference), situated 154.1 km from the Manaus urban area. We also analyzed BRAIN results in a Broken canopy, forested site north of Manaus (named T0t/TT34 in the GoAmazon reference), situated 60.1 km from the Manaus urban area. We also analyzed BRAIN results in the Manaus urban area (T1 site). Our analysis revealed that BRAIN clearly captured the seasonal patterns in T1 and T0a, especially for CO and $O_3$ concentrations. BRAIN estimates are consistent with concentrations in background areas of CO (up to ~ 350 ppb) and $NO_2$ (up to ~ 2.5 $\mu g.m^{-3}$) at T0t/TT34 (Figure 12) and T0a. However, it was not able to capture the seasonal profile at T0t/TT34. It shows that BRAIN could be an interesting alternative to fill the data gaps in background areas.

We have included the following sentences and figures in the manuscript:

*To analyze BRAIN's performance in background regions (with low anthropogenic influence), we extracted data from two forested sites in the Brazilian North region. We used as reference the sampling sites of the GoAmazon experiment (Martin et al., 2016) named T0a (forested site situated 154.1 km from the Manaus urban area) and T0t/TT34 (a Broken canopy, forested site situated 60.9 km from the Manaus). Sentinel-5P TROPOMI data spatially aligned to BRAIN resolution was also extracted for comparison. A buffer of 0.2° around the sites selected the data of CO, $O_3$, and $NO_2$ from both datasets. The results revealed that BRAIN captured the seasonal*

*profile at T0a (Figure 11), showing a moderate correlation with tropospheric column measurements of Sentinel-5P TROPOMI, especially for CO and $O_3$. BRAIN estimates are consistent with concentrations in background areas of CO (up to ~ 350 ppb), $O_3$ (up to ~ 0.04 ppm), and $NO_2$ (up to ~ 2.5 µg.m$^{-3}$) at T0t/TT34 (Figure 12) and T0a. However, it was not able to capture the seasonal profile at T0t/TT34.*

*We also analyzed BRAIN results in the Manaus urban area. We adopted the sampling site of the GoAmazon experiment (Martin et al., 2016) named T1 (INPA campus in Manaus). Compared with Sentinel-5P TROPOMI data, BRAIN fairly reproduced the temporal pattern of CO, $O_3$, and $NO_2$ in the T1 site (Figure 13).*

[Figure]

*Figure 11. Scatterplot and daily time series of CO (a), O₃ (b), and NO₂ (c) from BRAIN and Sentinel-5P TROPOMI at T0a (GoAmazon reference). Values extracted using a buffer of 0.2° around the station.*

[Figure]

*Figure 12. Scatterplot and daily time series of CO, O₃, and NO₂ from BRAIN and Sentinel-5P TROPOMI at T0t/TT34 (GoAmazon reference). Values extracted using a buffer of 0.2° around the station.*

[Figure]

Figure 13. Scatterplot and daily time series of CO (a), $O_3$ (b), and $NO_2$ (c) from BRAIN and Sentinel-5P TROPOMI at T1 (GoAmazon reference). Values extracted using a buffer of 0.2° around the station.

We also removed the "*background areas such as Manaus in Amazonas,*" from the manuscript.

**Comment #3:** In the Figure SM8, are the authors spatially averaging in the Manaus city? It is very hard to know the values on the dark red line (average). Can the authors provide a more

understandable figure?

Reply: Yes, the lineplot in Figure SM8 represents the spatial average and minimum-maximum of pixels within Manaus. The idea was providing a figure in each Brazilian capital. We provided a new Supplementary Material (SM19) only with averages of BRAIN data to improve the reader's understanding.

**Comment #4:** For the site TT34 located at the ZF2 ecological reserve (situated 60 km to the north-northwest of Manaus urban area in the Cuieiras Biological Reserve), based on the figure provided, BRAIN is not showing seasonal differences on the PM2.5 values. This site is located in a region with low anthropogenic influence. Why are the authors not comparing BRAIN outputs sampled at the forest site TT34 location?

Reply: We provided a new analysis comparing BRAIN and *Sentinel-5P TROPOMI* at TT34. Our analysis revealed that BRAIN results at TT34 are consistent with background concentrations. However, we haven't found meaningful correlation at TT34, which could be explained by the lower influence of anthropogenic sources.

We have included the following sentences and figures in the manuscript:

*To analyze BRAIN's performance in background regions (with low anthropogenic influence), we extracted data from two forested sites in the Brazilian North region. We used as reference the sampling sites of the GoAmazon experiment (Martin et al., 2016) named T0a (forested site situated 154.1 km from the Manaus urban area) and T0t/TT34 (a Broken canopy, forested site situated 60.9 km from the Manaus). Sentinel-5P TROPOMI data spatially aligned to BRAIN resolution was also extracted for comparison. A buffer of 0.2° around the sites selected the data of CO, O₃, and NO₂ from both datasets. The results revealed that BRAIN captured the seasonal profile at T0a (Figure 11), showing a moderate correlation with tropospheric column measurements of Sentinel-5P TROPOMI, especially for CO and O₃. BRAIN estimates are consistent with concentrations in background areas of CO (up to ~ 350 ppb), O₃ (up to ~ 0.04 ppm), and NO₂ (up to ~ 2.5 µg.m⁻³) at T0t/TT34 (Figure 12) and T0a. However, it was not able*

*to capture the seasonal profile at T0t/TT34.*

[Figure]

*Figure 12. Scatterplot and daily time series of CO, O₃, and NO₂ from BRAIN and Sentinel-5P TROPOMI at T0t/TT34 (GoAmazon reference). Values extracted using a buffer of 0.2° around the station.*

**Comment #5:** About the site in Porto Velho (PVH was located near Porto Velho city). The authors

should follow the same approach, compare the BRAIN results with the PVH site location, instead of spatially averaging a large area.

Reply: Ok, we now have compared with site locations at T0a, TT34, and T1. We will keep the spatially averaged figure in the manuscript since it brings a comparison between the databases (MERRA2, Sentinel-5P TROPOMI, and BRAIN).

**Comment #6:** Related with the PM2.5 analysis: The authors compared Artaxo et al., 2013 figure PM10 time series with BRAIN's PM2.5 time series in Porto Velho (Figure SM8). The seasonal change, with a peak in the dry season is shown in the BRAIN. Although it is important to emphasize the difference on the aerosol mode (particle size). Artaxo et al., shows fine and corse mode. BRAIN shows PM2.5 (fine particles) which is more similar with the coarse mode of the time series showed in Artaxo et al., 2013.

Reply: Ok. We included a new sentence in the discussion of the revised manuscript highlighting similarities with the coarse mode of the time series shown in Artaxo et al., 2013.

We have included the following sentences and figures in the manuscript:

*However, BRAIN $PM_{2.5}$ estimates are closer to the coarse mode of the time series, rather than the fine mode shown in Artaxo et al., (2013).*

**Comment #7:** Another point is related with the O3 analysis. BRAIN captured the absolute magnitude of O3 in September 2019 when compared with observed averaged data between 2009 to 2012 from Artaxo et al., 2013. It is important to mention the overestimation between the samples. BRAIN shows O3 2.7 times higher that the observation data (dry season). And a factor of 2 higher for the wet season of O3 values.

Reply: Ok. We mentioned the overestimation of BRAIN concerning the Artaxo et al., (2013) results. We also explain that Artaxo's campaing was conducted in 2013, while BRAIN in 2019. We have rephrase the sentence in the manuscript:

*BRAIN captures seasonal patterns and the absolute magnitude of $PM_{2.5}$ in the Northwest of the Amazonas state (near the Amazon Tall Tower Observatory -ATTO) presented by Artaxo et al., (2013). It shows that our database can reproduce the concentrations in background areas (far from highly urbanized centers). Comparing BRAIN with observations at heavy biomass burning impacted sites in south-western Amazonia (Porto Velho) (Artaxo et al., 2013) revealed that BRAIN can capture seasonal variations caused by wet and dry seasons and the magnitude of average and peak concentrations. However, BRAIN estimates are closer to the coarse mode of the time series, rather than the fine mode shown in Artaxo et al., 2013. Even though BRAIN has captured the $O_3$ pattern observed by Artaxo et al. (2013), the estimates are around 2.7 higher than the observations in the dry season and a factor of 2 higher for the wet season. It is worth mentioning that BRAIN uses 2019 data, while Artaxo et al. (2013) consists of a sampling campaign from 2008 to 2012.*

**Comment #8:** Figure 8 (Annual average concentration of CO and NO2 from BRAIN original resolution (a, c), SENTINEL/TROPOMI regridded to BRAIN resolution (b, d)). Can the authors convert the data to the same unit. Ex: all in PPB? In the same Figure, why is that so much of CO in the ocean region - b) Regrided SENTINEL/TROPOMI?

Reply: *Sentinel-5P TROPOMI* data are provided in $(mol.m^{-2})$ unit, representing the density of pollutants in the tropospheric column. For this reason, *Sentinel-5P TROPOMI* data cannot be straight-forward converted to surface concentration.

**Comment #9:** The idea of having a dataset as BRAIN is very needed specially due to the current lack of numerical simulated dataset available to work with air quality in Brazil. Also because not everyone that works with meteorological and/or air quality knows how to produce and work with atmospheric models. Nevertheless, the first step on evaluating and validating a modeling dataset in order to analyze regions beyond the AQS/aircraft is by comparing the modeling dataset against the most observational data available. After that, the modeling data set can be used to allows us to directly assess the regions where AQS are not available. I would expect most of the analysis in this paper to occur in areas with high AQS.

Reply: We appreciate the compliments.

**Comment #10:** Question: Which chem_opt option are you using in the namelist.input file?

Reply: If you mean chemical option of chemical transport model, we used Carbon Bond 06 mechanism. We provide all details of our simulation in the supplementary materials, including WRF namelists and CMAQ scripts.

References:

Martin, S.T., Artaxo, P., Machado, L.A.T., Manzi, A.O., Souza, R.A.F., Schumacher, C., Wang, J., Andreae, M.O., Barbosa, H.M.J., Fan, J., Fisch, G., Goldstein, A.H., Guenther, A., Jimenez, J.L., Pöschl, U., Silva Dias, M.A., Smith, J.N., and Wendisch, M. (2016). Introduction: Observations and Modeling of the Green Ocean Amazon (GoAmazon2014/5). Atmos. Chem. Phys., 16, 4785–4797, https://doi.org/10.5194/acp-16-4785-2016

---

## Author Response (AR3)

**Brazilian Atmospheric Inventories - BRAIN: A comprehensive database of air quality in Brazil**

MS No.: essd-2023-305

Dear editor,

We greatly appreciate the reviewer comments and suggestions, which are very constructive and have contributed to enhance the content of the revised manuscript. Please find below the reply to all the reviewer comments.

Best regards

Leonardo Hoinaski and coauthors

**Reply to comments of reviewer:**

**Comment #1:** The authors showed that BRAIN estimates are consistent with concentrations in background areas of CO (up to ~ 350 ppb), O3 (up to ~ 0.04 ppm) and NO2 (up to ~ 2.5 μg.m-3). I recommend the authors to mention in the paper the background values for O3 and CO known in the literature for the background regions such as ATTO site. The same approach needs to be followed for T1 site (Manaus city).

According with literature, O3 values to Atto site during wet season (March- April. 2013-2020) are around 7 ppbv ± 2 ppbv [Reference 01] and for TT34 (T0z site) O3 values were 8.5 ± 1.9 ppbv [Reference 02].

For background areas of CO in the Amazon region, especially during the GoAmazon experiment, I recommend the authors the see References [1, 3, 4, 5, 6 and 7]. Previous plume urban index with CO were already discussed for that region.

CO (up to ~ 350 ppbv), O3 (up to ~ 0.04 ppm) are not related with background values for ATTO site, unless a long-range transport event was influencing the site.

According with analysis t1 site, I recommend the authors to discuss about the observation values from CO and NOx previous showed in [Reference 08] and the values from Sentinel-5P TROPOMI and BRAIN data. It is important to point out some spatial-temporal differences when comparing three different kinds of data (model, satellite and ground station).

Reply: Thanks for the recommendation. We have mentioned the background values for $O_3$ and CO at the T0a and TT34 sites. We also added previous evidence about the CO values in the background region:

    *"BRAIN estimates are slightly higher than observed concentrations in background areas of CO, $O_3$, and $NO_2$ in TT34 (Figure 12) and T0a (Figure 11). While $O_3$ concentrations simulated by BRAIN range around 18 ppb (average in 2019) at the TT34 site, observed concentrations in 2013 (Artaxo et al., 2013) were around 8.5 ppb ± 1.9 ppb. In T0a, BRAIN simulated concentrations*

*around 16 ppb, overestimating the observations (7 ppb ± 2 ppb during the wet season from March to April 2013-2020) (Nascimento et al., 2022). Concerning CO, the concentrations simulated by BRAIN are slightly lower, ranging around 73 ppb (average) at TT34 against 130 ppb observed during the GoAmazon experiment from 2010 to 2011 (Artaxo et al., 2013). We emphasize that the BRAIN and GoAmazon datasets are reported in different periods and, consequently, influenced by different emissions rates. For instance, fire emissions have changed significantly since 2011 in Amazon (Copernicus, 2022; Naus et al., 2022)."*

[Figure]

**Figure 11. Scatterplot and daily time series of CO (a), O₃ (b), and NO₂ (c) from BRAIN and Sentinel-5P TROPOMI at T0a (GoAmazon reference). Values extracted using a buffer of 0.2° around the site.**

[Figure]

**Figure 12. Scatterplot and daily time series of CO, O₃, and NO₂ from BRAIN and Sentinel-5P TROPOMI at T0t/TT34 (GoAmazon reference). Values extracted using a buffer of 0.2° around the site.**

In addition, we have added new sentences in the revised manuscript to discuss the differences between observations and BRAIN simulations at the T1 site:

*"Rafee et al., (2017) reported mean concentrations of 88.7 ppb of NOx and 382.6 pbb of*

*CO in the Manaus urban area, while BRAIN reached 79 ppb and 99 ppb (maximum of 383 ppb), revealing an underestimation in this area. Again, the sampling campaign presented by Rafee et al. (2017) and BRAIN simulations uses different base year. Comparing BRAIN at T0a/TT34 (background sites) and T1 (urbanized), the database has reached consistent results with lower concentration levels in preserved areas.''*

[Figure]

**Figure 13. Scatterplot and daily time series of CO (a), O₃ (b), and NO₂ (c) from BRAIN and Sentinel-5P TROPOMI at T1 (GoAmazon reference). Values extracted using a buffer of 0.2° around the site.**

**References**

Artaxo, P., Rizzo, L.V., Brito, J.F., Barbosa, H.M., Arana, A., Sena, E.T., Cirino, G.G., Bastos, W., Martin, S.T., Andreae, M.O.: Atmospheric aerosols in Amazonia and land use change: from natural biogenic to biomass burning conditions, Faraday Discuss., 165, 203–235, https://doi.org/10.1039/C3FD00052D, 2013.

Copernicus, Wildfires: Amazonas records highest emissions in 20 years, https://atmosphere.copernicus.eu/wildfires-amazonas-records-highest-emissions-20-years (last access Mar 2024), 2022.

Nascimento, J.P., Barbosa, H.M.J., Banducci, A.L., Rizzo, L.V., Vara-Vela, A.L., Meller, B.B., Gomes, H., Cezar, A., Franco, M.A., Ponczek, M., Wolff S., Bela M.M., Artaxo P.: Major Regional-Scale Production of $O_3$ and Secondary Organic Aerosol in Remote Amazon Regions from the Dynamics and Photochemistry of Urban and Forest Emissions, Environ. Sci. Technol., 56, 9924–9935, https://doi.org/10.1021/acs.est.2c01358, 2022.

Naus, S., Domingues, L.G., Krol, M., Luijkx, I.T., Gatti, L.V., Miller, J.B., Gloor, E., Basu, S., Correia, C., Koren, G., Worden, H.M., Flemming, J., Pétron, G., Peters, W.: Sixteen years of MOPITT satellite data strongly constrain Amazon CO fire emissions, Atmos. Chem. Phys., 22, 14735–14750, https://doi.org/10.5194/acp-22-14735-2022, 2022.

Rafee, S.A.A., Martins, L.D., Kawashima, A.B., Almeida, D.S., Morais, M. V.B., Souza, R.V.A., Oliveira, M.B.L., Souza, R.A.F., Medeiros, A.S.S., Urbina, V., Freitas, E.D., Martin, S.T., Martins, J.A.: Contributions of mobile, stationary and biogenic sources to air pollution in the Amazon rainforest: a numerical study with the WRF-Chem model, Atmos. Chem. Phys., 17, 7977–7995, https://doi.org/10.5194/acp-17-7977-2017, 2017.

**Comment #2:** In Figure 11 and others, the units of the BRAIN result and Sentinel-5P TROPOMI should be the same to facilitate comparison between the data sets. If you cannot convert the Sentinel/TROPOMI data into PPB, integrate the BRAIN column and convert it to mol m^-2. Also, in these figures there is a hidden offset that is not mentioned, eg in figure 11 a on the right, the bottom of both y axes are not zero. This further increases the difficulty of comparing the datasets.

Reply: Unfortunately, we provide only surface concentrations in BRAIN, therefore, we cannot integrate it vertically. Our choice was to provide multiple species rather than layers. Files with multiple species and vertical layers are too large to store in long-term repositories. Also, keeping these files along modeling would require a super large storage capacity. The comparison of BRAIN

and SENTINEL/TROPOMI shows the similarities between the variables' variances.

Cloud coverage influences Sentinel measurements, reproducing negative values in the time series. Our previous idea was to compare the raw versions of the datasets. We agree that removing these low-quality values in the SENTINEL/TROPOMI products could reproduce a more consistent analysis. We revised all figures derived from SENTINEL/TROPOMI data, removing values smaller than 0.

[Figure]

**Figure 13. Scatterplot and daily time series of CO (a), O₃ (b), and NO₂ (c) from BRAIN and Sentinel-5P TROPOMI at T1**

(GoAmazon reference). Values extracted using a buffer of 0.2° around the site.